# BRIDGE: BOOTSTRAPPING TEXT TO GUIDE TIME-SERIES GENERATION VIA MULTI-AGENT ITERATIVE OPTIMISATION AND DIFFUSION MODELLING

## ABSTRACT

Time-series Generation (TSG) is an impactful research direction, as generating realistic sequences can be used to create educational materials, in simulations and for counterfactual analysis in decision making. It has further the potential to alleviate the resource bottleneck that arises from a lack of diverse time-series data required to train large time-series foundational models. However, most existing TSG models are typically designed to generate data from a specified domain, which is due to the large divergence in patterns between different real-world TS domains. In this paper, we argue that text can provide semantic information (including cross-domain background knowledge and instance temporal patterns) to improve the generalisation of TSG. To do so, we introduce "Text Guided Time Series Generation" ($TG^2$)—the task of generating realistic time series from handful of example time series paired with their textual description. We further present a Self-Refine-based Multi-Agent LLM framework to synthesise a realistic benchmark for $TG^2$ and show that the collected text descriptions are both realistic and useful for time-series generation. We develop a first strong baseline for the $TG^2$, BRIDGE, which utilises LLMs and diffusion models to generate time series which encode semantic information as cross-domain condition. Our experimental results demonstrate that BRIDGE significantly outperforms existing time-series generation baselines on 10 out of 12 datasets, resulting in data distributions that are more closely aligned to target domains. Using the generated data for training positively impacts the performance of time series forecasting models, effectively addressing training data limitations. This work bridges the gap between LLMs and time series analysis, introducing natural language to help the TSG and its applications.

## 1 INTRODUCTION

The generation of time series (TS) data is an important task in various domains, including finance (Sezer et al., 2020), healthcare (Hong & Chun, 2023), meteorology and environmental science (Hasnain et al., 2022). For example, realistic synthetic medical electrocardiogram (ECG) patterns can be used to train medical residents (Hong & Chun, 2023), while simulating regional electricity usage can be used for stress testing the power grid (Westgaard et al., 2021). Previous methods like TimeGAN and TCGAN (Huang & Deng, 2023) utilise GANs to produce realistic TS, showing remarkable performance even with limited labeled data. Similarly, VAE-based approaches enable decoupling the mapping process from standard VAE training, allowing for precise control over generated outputs (Bao et al., 2024). However, such models are confined to generating single-domain data. In contrast, generating TS representations from unseen domains during training introduces additional complexities, as real data resources are often scarce, private, and highly valuable, while TS patterns and scales vary significantly across different domains. This stands in stark contrast to the domains of NLP and CV, where the availability of large-scale datasets has led to foundational Large Language Models (LLMs), which have demonstrated strong generalization and reasoning abilities (Brown et al., 2020; Mirchandani et al., 2023), and have shown efficient utilization of data (Wang et al., 2024), even in few-shot or zero-shot scenarios. In particular, their demonstrated ability to generate images (Zheng et al., 2023) and videos (Liu et al., 2024e) from text prompts, creating a timely opportunity to extend these capabilities to other modalities, such as time series. Leveraging text as a source of cross-domain information for TSG via LLMs could facilitate the capture of complex patterns and semantic relationships, akin to their application in other domains.

Recent research has explored two approaches to leveraging LLMs for time series analysis: adapting existing LLMs to handle time series data (TS-for-LLM) and developing specialized LLMs for time series from scratch (LLM-for-TS). The TS-for-LLM approach aims to utilize LLMs' semantic capabilities by representing time series as word embeddings, requiring minimal training and data (Wang et al., 2022; Ye et al., 2024). Methods include aligning word and time-series embeddings through clustering (Pan et al., 2024) and contrastive learning (Sun et al., 2023). However, this approach faces challenges in accurately representing continuous time series data with discrete vocabularies and may not necessarily require LLMs (Tan et al., 2024). The LLM-for-TS approach seeks a more fundamental solution by pre-training models on time series data, as exemplified by TimesFM (Das et al., 2023) and Chronos (Ansari et al., 2024). While these methods have shown promising results, they primarily focus on building foundational models for time series forecasting. In contrast, the challenge of cross-domain time series generation, particularly leveraging textual information to guide and enhance the generation process, remains underexplored. In addition, the limited availability of time series data compared to NLP and CV domains poses a significant challenge in developing models with emergent abilities similar to traditional LLMs, making it difficult to consistently meet the data requirements for such approaches.

We argue that using text to assist in cross-domain TS generation can help overcome the data scarcity issues inherent to the TS domain, as the knowledge provided can be transferred to other domains (Shang et al., 2021). Side-stepping the issues associated with TS-for-LLM and LLM-for-TS, we do not directly input TS data into the LLMs or pre-train LLMs on TS. Instead, we take an intermediate step by learning Text-to-TS prototypes (also known as bases (Harpham & Dawson, 2006)) that serve as basic elements to construct soft prompts. These prototypes capture underlying temporal patterns, such as trends, seasonalities, and semantic information for domains, which are used to generate TS data with a diffusion model. During training, the proposed model uses both text descriptions and TS samples as input, employing a prototype assignment module to create tailored "prompts" for each sample. During sampling, texts and few-shot samples serve as context to generate "prompts", which condition TS generation in a process akin to instruction tuning (Zhang et al., 2023). Here, LLMs act as assistants rather than generators, leveraging the accessibility of text while avoiding the limitations of conventional LLM-based TS methods. In this setup, the proposed model achieved state-of-the-art performance on the majority of datasets and demonstrated strong robustness in few-shot learning scenarios, particularly on unseen datasets.

The lack of resources for TS is particularly pronounced for $TG^2$ tasks, which poses a significant challenge in validating our proposed approach. This likely stems from the difficulty in precisely describing TS with words (Yang & Lee, 2009; Liu et al., 2024a). The nature of automatically finding textual descriptions for TS is akin to prompt optimisation for LLMs, where prompt variations greatly impact performance (T et al., 2024). Although automated prompt generation methods like random search (Zhou et al., 2023b), genetic algorithms (Liu et al., 2024c), and reinforcement learning (Guo et al., 2024a) have been developed, they were not applied for $TG^2$. To address this gap, we leverage on the recent advancements in LLM-based multi-agent systems for complex problem-solving (Guo et al., 2024b) and propose a role-based LLM collaborative multi-agent framework to generate a high-quality benchmark for $TG^2$. The experimental results highlight the significance of the proposed framework. Compared to the original text, the revised text achieves at least a 15% performance boost. Additionally, multi-agent collaboration systems provide more comprehensive outputs compared to the straightforward generated text.

To summarise, this paper makes the following novel contributions: **First**, We propose a multi-agent framework to create a text guided time series generation $TG^2$ benchmark. Our numeric experiments show that the descriptions provide helpful information for time-series models. **Second** with this benchmark, we analyse the impact of different types of time-series descriptions, which advances the understanding of how LLM can be used to assist time series prediction and generation in a zero-shot setting. **Third**, we propose BRIDGE, a novel text-based time series generation framework via LLMs and diffusion. The proposed method outperforms all baselines on 10 out of 12 datasets and achieves the best performance on data from unseen domains in a few-shot setting, demonstrating strong cross-domain generalization. **Finally**, we show that the BRIDGE effectively addresses the lack of time-series resources as forecasting models trained on synthetic data perform similarly compared to when trained on real data.

## 2 RELATED WORK

**Large Language Models for Time Series:** Recent studies explore LLMs for time series (TS) analysis. Some, like Das et al. (2023), pre-train models from scratch, while others, such as Chronos (Ansari et al., 2024), tokenize TS data to leverage NLP techniques. These methods achieve strong performance but require significant computational resources, limiting scalability. Alternative approaches align LLMs with TS embeddings, as seen in EEG-to-Text (Wang & Ji, 2022) and GPT4TS (Zhou et al., 2023a). Enhancements include trend decomposition (TEMPO (Cao et al., 2024)), two-stage fine-tuning (LLM4TS (Chang et al., 2023)), and specialized embeddings or architectures (e.g., UniTime (Liu et al., 2024d), GATGPT (Chen et al., 2023), ST-LLM (Liu et al., 2024b)). Time-LLM and Lag-Llama apply LLaMA for TS tasks (Jin et al., 2023; Rasul et al., 2023). Despite progress, challenges remain in bridging the gap between discrete text and continuous TS data.

**Time Series Generation:** Traditional TS generation has used various generative models to capture temporal structure. GANs were among the first, using supervised and adversarial objectives to encourage temporal coherence, as in TimeGAN (Yoon et al., 2019). VAEs adapt to TS by adding decoder structures for trend and seasonal components (Desai et al., 2021). Advances include vector quantization with bidirectional transformers, enhancing temporal consistency (Lee et al., 2023), and mixed models combining GANs, normalizing flows, and ODEs for complex patterns (Jeon et al., 2022). Denoising diffusion models (DDPMs) generate TS by reversing a noise-added Markov process and support conditional generation, although current models lack domain-specific conditional details (Sohl-Dickstein et al., 2015; Ho et al., 2020).

## 3 ITERATIVE OPTIMISATION: MULTI-AGENT COLLABORATION TO REFINE A TEXT DESCRIPTION

In this section, we discuss the experimental need for iterative optimization and the detailed architecture of the proposed multi-agent collaboration framework. Specifically, we first validate the challenges of using LLMs as TS generators and directly optimising them. Then, we discuss the hierarchical strategy of our multi-agent collaboration framework, which includes two main components: the first divides multiple agents into two teams that independently execute tasks, and the second employs an existing model (Liu et al., 2024a) for testing and providing feedback, enabling the teams to iterate further on the results.

### 3.1 IS DIRECT OPTIMIZATION FEASIBLE?

We first explored whether it is feasible to directly use human-readable text descriptions to prompt LLMs to improve performance. During the experiment, the LLMs still struggles to grasp the overall trend of gradual increase, even if we adopt Seasonal-trend decomposition using Loess (STL) (Cleveland et al., 1990) to further decompose the TS, as shown in Appendix A.1. This is similar to the findings of Merrill et al. (2024), where LLMs still require additional assistance, such as chain-of-thought reasoning or input in a format that the model can comprehend, to be effective.

### 3.2 MULTI-AGENT COLLABORATION SYSTEM FOR ITERATIVE OPTIMISATION

**Step 1 Building the Initial Description:** As noted in previous work (Merrill et al., 2024), generating fine-grained text descriptions remains a challenging task due to the limited availability of extensive data resources. To address this, we take an intermediate step by narrating key information related to time series in a standardized text format. Starting with a variety of initial queries, we first identify and collect articles, papers, news, and reports that describe data similar to time series. While direct search for relevant content is feasible, it is constrained by a maximum of $K$ titles relevant to the query keyword. To overcome this, we aim to gather relevant candidates based on content similarity. For instance, a simple search for *"time series generation"* might return its definition, but a reasoning-enabled agent can plan what types of articles are more likely to contain relevant content, thereby diversifying the search results. Therefore, we propose a single-agent framework inspired by ReAct (Yao et al., 2023), which prompts LLMs to generate dynamic reasoning traces for collecting candidates and actions to interact with external environments (e.g., Google, Wikipedia) in an interleaved manner (Madaan et al., 2023) (Framework pipeline can be find in Appendix A.2). The agent analyzes and decomposes the query into sub-questions, using external tools to answer

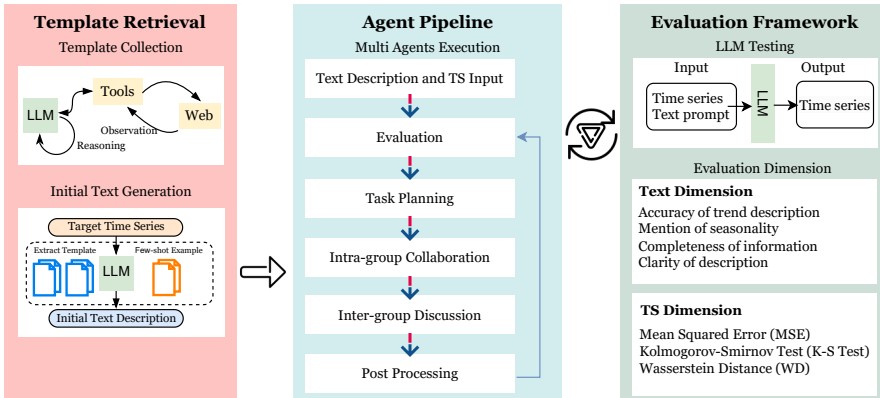

Figure 1: The pipeline of the proposed multi-agent collaborative framework consists of three main stages: *(i)* Initial Text Generation: Templates are collected and extracted from OpenWeb to generate an initial text description. *(ii)* Multi-Agent Execution: Agents collaborate on task planning, intra-group coordination, and inter-group discussions to generate and refine text descriptions. *(iii)* Evaluation: The generated outputs are evaluated using statistical metrics and text quality dimensions, such as clarity and accuracy.al metrics and text quality dimensions such as clarity and accuracy.

each sub-question iteratively until all are addressed. Afterward, another LLM extracts general time series templates from the collected corpus. These templates contain descriptions of general trends, seasonality, and background information. The detailed example can be viewed in Appendix A.3.

**Step 2 Evaluating the Input/Revised Text:** In this step, our goal is to provide the system with the ability to evaluate text generations and provide feedback. Since the overall objective is to utilize text to assist and guide, we define the testing phase as a TS forecasting task with accompanying text—better text should lead to better TS forcasting performance. Specifically, the input consists of a TS along with its textual description as conditions, and the goal is to forecast future TS. This arises from the intuition that historical TS can serve as supplementary contextual information, reducing the complexity of the generation process and providing a constraining effect.This approach maximizes the evaluation of the text's impact while minimizing the influence of the time series itself. A straightforward approach would be to use existing forecasting models fine-tuned on TS as the backbone. However, these models generally input and output data in time series format (Zhou et al., 2023a). Recent work has shown that advanced prompting strategies can leverage the capabilities of large language models (LLMs) for zero-shot TS forecasting (Gruver et al., 2023). Thus, we employed LSTPrompt as our evaluation backbone, which prompts off-the-shelf LLMs with chain-of-thought (CoT) reasoning (Wei et al., 2022), enabling the integration of text as an additional input modality (Liu et al., 2024a). The key to our refined framework lies in the definition of the evaluation dimension, which directly influences the agent's ability to correct text-time series pair errors and provide high-quality feedback. On one hand, we define evaluation criteria that align with the modal characteristics of both text and TS, allowing the agent to consider both simultaneously in order to correct the data. On the other hand, we also allow the agent to propose more suitable evaluation metrics. For the detailed initial evaluation criteria and definitions, refer to Appendix A.4.

**Step 3 Iteratively Refining the Text Description:** Initial descriptions may be coarse or contain errors. To generate text that is optimized for LLM processing while remaining suitable for human understanding, we propose a multi-agent collaboration system that simulates the iterative refinement process of a team of human prompt engineers, leveraging the demonstrated capability of LLMs to improve their own outputs (Zhang et al., 2024). As illustrated in Figure 1, the system operates through three stages: **Stage 1 Task Planning** (assigning tasks and monitoring progress), **Stage 3 Inter-group Discussion** (independent teams iteratively refining outputs), and inter-team discussion (collaborative consensus building). Refined outputs are validated and incorporated into a formal dataset, while templates are added to a general library for future use. More Detail about system structure, output and sample example can be find at Appendix A.5, Appendix A.6 and Appendix A.7.

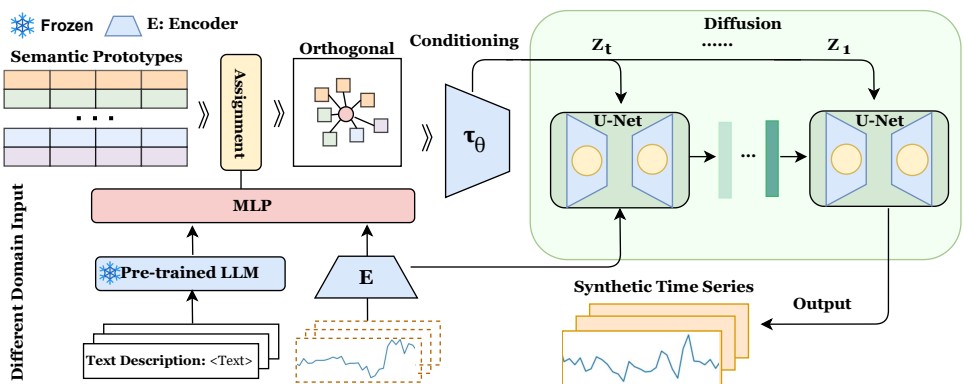

Figure 2: Overview of the proposed framework BRIDGE. The input text is processed into word embeddings and, along with the corresponding time series, is fed into an assignment module to orthogonalize with semantic prototypes. This is then used as a condition for the diffusion model. When no text is provided, the model can also function as an unconditional diffusion model.

## 4 DIFFUSION-BASED TEXT ASSIST-TIME SERIES GENERATION MODEL

In this section, we describe our framework for TG$^2$ in detail. As can be seen from Figure 2, the proposed framework first converts and fuses different modality inputs into embeddings and then assigns them with the corresponding semantic prototypes. Then, a diffusion model is conditioned on the semantic prototypes and generate TS samples using few Text-TS pairs as demonstrations.

### 4.1 PROBLEM FORMULATION

Consider training time series datasets $D_s$ gathered from a specific data source $s \in S$. Given the dataset $D = \{X^i_{1:\tau}\}^N_{i=1}$ of $N$ samples of time-series, covering a period of $\tau$ time steps, the goal of TSG is to learn a model distribution $p_\theta(X)$ that approximates the data distribution $q_D(X)$. In this work, we aim to enable few-shot learning in cross-domain TSG, analogous to NLP where new data is generated based on task descriptions and examples, even in an unseen domain.

**Definition 1** (Text-Guided Time Series Generation (TG$^2$)). *Let $X_s$ represent an observation from domain $s$, which may originate from the training sets $S$ or from an unseen domain. Given its corresponding text description $U_s$, our objective is to learn the conditional distribution $p_\theta(X|X_s, U_s)$ to approximate the true data distribution $q_{D_s}(X)$, without providing a domain label $s$.*

### 4.2 SEMANTIC PROTOTYPING

During model training, the data utilized covers multiple domains, necessitating the model to possess robust generalization capabilities to learn the distinct distribution characteristics of different domains. However, during inference, domain labels are not provided; instead, only sample data and its descriptions are available. Consequently, the model must infer the potential domain to which the sample data belongs—if it aligns with any domains encountered during training—or deduce the characteristics of the domain if it is novel. This highlights the need for models capable of effectively analyzing and generalizing across diverse data distributions. To tackle this issue, we propose using semantic prototypes to encode knowledge from different perspectives and employ adaptive prototype allocation to associate features with time series, referred to as Bases (Ni et al., 2023). Bases represent a small set of fundamental features extracted from time series data. Each basis encapsulates certain core attributes of time series. While each observed sequence may exhibit different realisations of these attributes, the bases should come from the same pool, meaning that every time series in the dataset can be reconstructed by weighting these basis. Thus, we can utilize these bases as common knowledge to bridge across domains. We define a set of latent arrays as **prototype** $P \in \mathbb{R}^{n_p \times d}$ to represent domain-agnostic time-series commonsense. The prototypes $P$ are initially set with random orthogonal vectors and then fixed.

### 4.3 SEMANTIC PROTOTYPE ASSIGNMENT

Although the same set of prototypes is used across different instances, the degree to which each prototype explains different instances varies. To address this, we assign prototypes to each time series and text description pair, which serves as a condition for the generation model. For each input sequence $x$ (comprising both the time series and text embeddings), a weight vector is generated, the dimension of which corresponds to the number of prototypes. This is achieved via a feature extractor $\phi$. Each element of the vector $\phi(x)_i$ reflects the contribution of each prototype unit $p_i$ in the prototype set $P$, and these weights modify the attention mechanism used during generation. As a result, the model is conditioned on the assigned weighted prototypes. The weights are applied through an attention mask $m$, which operates on the attention weights for prototypes. To ensure sparsity, we discard prototype units that are assigned with negative weights by setting their attention weights to zero. Formally, the prototype assignments are transformed into attention mask $m$ as follows:

$$m = \phi(x_0, t_0) - I_{\phi(x_0,t_0)\leq 0} \cdot \infty \tag{1}$$

where $\phi(x,t) \in \mathbb{R}^{N \times d}$ is the output from the feature extraction layer that processes both time series and text embeddings. $I_{\phi(x,t)\leq 0}$ is an indicator function that zeroes out negative weights, ensuring that only retains non-negative values.

### 4.4 SEMANTIC PROTOTYPE ALIGNMENT

To condition the denoising diffusion process, we adapt the denoising objective using $c$ as a condition, influencing the model's intermediate layers through cross-attention. This ensures that the generated time series aligns with the specified conditional instruction. To achieve this, we aim to align the condition and semantic prototypes during the training phase. A set number of query embeddings are allocated to both the text and time series as input. These queries interact with each semantic prototype through cross-attention layers (inserted every other transformer block $z$). We initialise the weights of the cross-attention layers randomly and update them during training. Specifically, we apply cross-attention to the feature representations using the following equations:

$$Q = W_Q \cdot cz, \quad K = W_K \cdot \mathbf{P}, \quad V = W_V \cdot \mathbf{P} \tag{2}$$

where

$$z = \text{FF}\left(\text{softmax}\left(\frac{Q(K)^T}{\sqrt{d}} + m\right) \cdot V\right) \tag{3}$$

Here, $z \in \mathbb{R}^{N \times d}$ denotes the output from the attention block. $W_Q, W_K, W_V \in \mathbb{R}^{d \times d}$ are learnable projection matrices applied on the sequence dimension. The attention output $z_{\text{final}}$ is passed to another feedforward network to produce the final output $\hat{\epsilon} = \text{FF}(z_{\text{final}})$.

### 4.5 DIFFUSION BASED $TG^2$ MODEL

As shown in Figure 2, instead of training separate models for each dataset, we propose a unified training approach that leverages data from multiple domains simultaneously. While each dataset may represent only a small fraction of the overall data distribution, this strategy allows the model to capture a broader range of patterns by sharing information across domains. During generation, both the time series and text data are fed into the model as joint embeddings, where we use an MLP to project the text embeddings from the LLM into the same dimensional space as the time series embeddings. The projected text is then prepended to the input embeddings, functioning as soft prompts that condition the diffusion model based on the contextual information extracted from the text. By constructing the conditioning input in this manner, the model generates samples that adhere to the selected domain while avoiding being constrained by the general temporal patterns exhibited in the selected samples. When the number of expected generated samples exceeds larger than the number of input sample, we employ a strategy of repeatedly generating with each assignment in selected samples until the number of expected samples is satisfied.

## 5 EXPERIMENT SETTING

Broadly speaking, our aim is to investigate the feasibility using text to guide TS generation. Specifically, we ask: *(i) What kinds of strategies are more efficient for the proposed multi-agent system?*

*(ii)* What types of text is helpful to generate time series? *(iii)* Does the proposed BRIDGE achieve competitive performance in TSG compared to SOTA TSG models? *(iv)* Can synthetic data be used to help improve the performance of TS task? *(v)* What is the role of text? Is it helpful? *(vi)* What is the impact of different configurations of prototypes? To answer questions *(i)* and *(ii)*, we conducted experiments on SOTA models that allow text input. For questions *(iii)* and *(iv)*, we evaluated the generation quality on the same dataset settings. Finally, For questions *(v)* and *(vi)*, we performed ablation experiments.

**Baseline Introduction** We compare to SOTA TS methods for both TS generation and forecasting tasks. For generation, we explore the performance of BRIDGE by comparing with conditional (`TimeVQVAE`, Lee et al. 2023) and unconditional approaches (`TimegGAN`, Yoon et al. 2019; `GT-GAN`, Jeon et al. 2022; `TimeVAE`, Desai et al. 2021, `DDPM` (Ho et al., 2020) ). For forecasting, our goal is to establish the realism of synthetic data. Here, we compare the performance of `Time-LLM` (Jin et al., 2023), LLM4TS (Chang et al., 2023) and TEMPO (Cao et al., 2024), GPT4TS (Zhou et al., 2023a). Detailed descriptions can be found in Appendix C.2. More details about Experiment Setup and Implementation can be found at Appendix D and Appendix E.

**Datasets** We evaluate the effectiveness of BRIDGE on 12 uni-variate datasets including Electricity, Solar, Wind, Traffic, Taxi, Pedestrian, Air, Temperature, Rain, NN5, Fred-MD, Exchange. These datasets have been widely used as benchmark datasets for TS generation tasks. We use ILI and M4 (Makridakis et al., 2018) datasets for forecasting task. Details of these datasets are in Appendix F.2.

**Evaluation Metrics** For time series generation, we measure Marginal distribution difference (MDD) and Kullback-Leibler divergence (K-L) to quantify the distribution difference between real and synthetic data. The detail are reported in Appendix G Following established evaluation protocols (Wu et al., 2023), we measure the Mean Square Error (MSE) and Mean Absolute Error (MAE) for long-term forecasting. For short-term forecasting on the M4 benchmark, we adopt the Symmetric Mean Absolute Percentage Error (SMAPE), Mean Absolute Scaled Error (MASE), and Overall Weighted Average (OWA) as evaluation metrics (Oreshkin et al., 2020).

# 6 RESULTS AND ANALYSIS

## 6.1 LLM-BASED AGENT STRATEGIES ANALYSIS

We first verified what strategy is most useful. Macro refers to a single team executing high-level information adjustments, while Micro focuses on details. Multiple teams indicate collaboration between two teams to complete the task. Overall, collaboration among multiple teams outperforms any single-team strategy, indicating that combining different strategies leads to more comprehensive and appropriate textual outputs. From a strategic perspec-

Table 1: The impact of different strategies of agent system. The ablations experiment on zero-short setting (MAE reported).

| Policy | Airpassenger | | Sunspots | |
|---|---|---|---|---|
| | LLMTime | LSTPrompt | LLMTime | LSTPrompt |
| *Multi* | 40.94 | 12.39 | 48.64 | 42.37 |
| *Single (Micro)* | 44.27 | 14.22 | 56.80 | 45.70 |
| *Single (Macro)* | 42.57 | 13.83 | 54.51 | 45.01 |

tive, the macro single-team approach performs better than the micro single-team approach, suggesting that overly detailed textual descriptions are still challenging to utilize effectively at this stage. Both teams chose to include statistical information, aligning with previous work that these factors most intuitively provide valuable insights, detail example can be find in Appendix A.7.

## 6.2 WHAT KIND OF TEXT IS USEFUL?

**Conciseness leads to better performance:** Table 2 shows that concise text inputs outperform overly detailed ones, which can mislead the model. This is particularly evident in the case of "w/o instance context", where the MAE improves by 1.6 (compared to "Initial text") on the Air-Passenger dataset, indicating that generating text that fully aligns with human preferences remains a challenging task. Notably, when it comes to longer sequence length, the context provides more useful information (48.64 vs 59.91 on Sunspots). **Clearly specifying the length of the prediction/generation can make the model's performance more stable.** This can be seen from the performance of "w/o statistics". After providing a clear sequence length and statistical values, the model's performance improves. **Background information helps the model.** Similar to the findings of other works (Jin et al., 2023; Merrill et al., 2024), backround infor-

mation can significantly improve the model's performance. This is likely because retrieving the pre-trained knowledge from the LLM's can offer additional contextual information as support.

**Direct pattern descriptions are more effective than detailed trend descriptions**. As mentioned in Section 3.1, when attempting to decompose the TS into seasonal, trend, and residual components, the model's performance did not show significant improvement. After multiple iterations, the most effective method was to provide the overall upward/downward trends and explicitly identify the top $k$ extreme points.

| Variant | AirPassenger | | Sunspots | |
|---|---|---|---|---|
| | LLMTime | LSTPrompt | LLMTime | LSTPrompt |
| *Initial text* | 49.36 | 15.12 | 59.88 | 49.71 |
| *Revised text* | 40.94 | 12.39 | 48.64 | 42.37 |
| *w/o Instance Context* | 41.96 | 13.54 | 54.33 | 44.23 |
| *w/o Background* | 44.63 | 14.77 | 56.81 | 46.07 |
| *w/o Statistical Context* | 44.01 | 13.41 | 54.24 | 47.12 |
| *w/o Pattern* | 44.36 | 14.52 | 55.16 | 46.84 |
| *w/o Pattern+Statistic* | 44.30 | 14.27 | 56.89 | 45.65 |
| *Baseline* | 45.75 | 15.00 | 59.91 | 47.59 |

Table 2: Ablation study for zero-shot time series forecasting (MAE reported).

## 6.3 TIME SERIES GENERATION QUALITY ASSESSMENT

Table 3: Generation result on various univarite datasets. Marginal distribution distance scores (MDD) and K-L divergence (K-L) are reported. A lower value indicates better performance. Best results are highlighted in Red and the second best results are Blue.

| | Dataset | Bridge | Bridge w/o Text | TimeVQVAE | TimeGAN | GT-GAN | TimeVAE | DDPM |
|---|---|---|---|---|---|---|---|---|
| Marginal Distribution Distance | Electricity | $0.206 \pm 0.050$ | $0.252 \pm 0.047$ | $2.763 \pm 0.088$ | $2.443 \pm 0.765$ | $2.026 \pm 0.280$ | $3.306 \pm 0.044$ | $1.045 \pm 0.385$ |
| | Solar | $375.533 \pm 10.110$ | $375.908 \pm 10.230$ | $466.174 \pm 0.145$ | $460.810 \pm 14.078$ | $476.196 \pm 17.041$ | $365.906 \pm 6.365$ | $379.256 \pm 0.100$ |
| | Wind | $0.365 \pm 0.062$ | $0.435 \pm 0.076$ | $0.777 \pm 0.028$ | $1.115 \pm 0.159$ | $0.706 \pm 0.106$ | $0.943 \pm 0.008$ | $0.620 \pm 0.140$ |
| | Traffic | $1.168 \pm 0.020$ | $1.209 \pm 0.011$ | $1.170 \pm 0.028$ | $1.733 \pm 0.137$ | $1.311 \pm 0.032$ | $0.984 \pm 0.012$ | $1.505 \pm 0.058$ |
| | Taxi | $0.591 \pm 0.051$ | $0.812 \pm 0.040$ | $0.534 \pm 0.032$ | $1.278 \pm 0.168$ | $1.118 \pm 0.157$ | $0.697 \pm 0.007$ | $1.214 \pm 0.186$ |
| | Pedestrian | $1.240 \pm 0.047$ | $1.075 \pm 0.045$ | $1.625 \pm 0.060$ | $1.574 \pm 0.290$ | $1.559 \pm 0.117$ | $0.777 \pm 0.012$ | $1.640 \pm 0.130$ |
| | Air | $0.633 \pm 0.045$ | $1.105 \pm 0.115$ | $0.338 \pm 0.012$ | $2.089 \pm 0.618$ | $2.828 \pm 0.172$ | $1.369 \pm 0.040$ | $1.481 \pm 0.057$ |
| | Temperature | $0.552 \pm 0.025$ | $0.618 \pm 0.029$ | $0.943 \pm 0.035$ | $1.164 \pm 0.110$ | $1.165 \pm 0.072$ | $2.044 \pm 0.024$ | $0.809 \pm 0.147$ |
| | Rain | $9.554 \pm 0.030$ | $9.890 \pm 0.055$ | $9.243 \pm 0.122$ | $10.937 \pm 4.039$ | $6.473 \pm 1.207$ | $9.134 \pm 0.477$ | $9.812 \pm 0.566$ |
| | NN5 | $1.340 \pm 0.032$ | $1.891 \pm 0.040$ | $1.424 \pm 0.043$ | $2.758 \pm 0.142$ | $2.121 \pm 0.094$ | $2.871 \pm 0.045$ | $1.498 \pm 0.245$ |
| | Fred-MD | $0.388 \pm 0.082$ | $0.614 \pm 0.014$ | $2.932 \pm 0.133$ | $4.028 \pm 0.130$ | $4.026 \pm 0.087$ | $2.902 \pm 0.215$ | $1.127 \pm 0.043$ |
| | Exchange | $0.392 \pm 0.048$ | $0.489 \pm 0.033$ | $0.993 \pm 0.058$ | $1.553 \pm 0.122$ | $1.355 \pm 0.072$ | $1.331 \pm 0.042$ | $0.631 \pm 0.584$ |
| K-L Divergence | Electricity | $0.006 \pm 0.003$ | $0.008 \pm 0.002$ | $0.185 \pm 0.018$ | $0.395 \pm 0.121$ | $0.415 \pm 0.040$ | $0.580 \pm 0.005$ | $0.014 \pm 0.002$ |
| | Solar | $0.032 \pm 0.004$ | $0.046 \pm 0.002$ | $0.726 \pm 0.043$ | $0.889 \pm 0.288$ | $0.102 \pm 0.045$ | $0.201 \pm 0.008$ | $0.291 \pm 0.069$ |
| | Wind | $0.112 \pm 0.032$ | $0.144 \pm 0.036$ | $0.493 \pm 0.081$ | $4.528 \pm 1.743$ | $0.511 \pm 0.129$ | $0.553 \pm 0.014$ | $0.412 \pm 0.144$ |
| | Traffic | $0.022 \pm 0.006$ | $0.055 \pm 0.005$ | $0.145 \pm 0.015$ | $2.134 \pm 0.952$ | $1.108 \pm 0.171$ | $0.212 \pm 0.006$ | $0.255 \pm 0.154$ |
| | Taxi | $0.083 \pm 0.016$ | $0.192 \pm 0.013$ | $0.100 \pm 0.014$ | $1.160 \pm 0.651$ | $0.663 \pm 0.127$ | $0.120 \pm 0.005$ | $0.348 \pm 0.147$ |
| | Pedestrian | $0.072 \pm 0.007$ | $0.040 \pm 0.004$ | $0.275 \pm 0.021$ | $0.881 \pm 0.436$ | $0.347 \pm 0.085$ | $0.052 \pm 0.010$ | $0.289 \pm 0.164$ |
| | Air | $0.032 \pm 0.012$ | $0.106 \pm 0.010$ | $0.017 \pm 0.004$ | $0.588 \pm 0.369$ | $0.506 \pm 0.091$ | $0.176 \pm 0.016$ | $0.213 \pm 0.085$ |
| | Temperature | $0.884 \pm 0.022$ | $0.085 \pm 0.015$ | $0.980 \pm 0.190$ | $8.775 \pm 2.511$ | $2.177 \pm 0.323$ | $1.910 \pm 0.076$ | $0.511 \pm 0.129$ |
| | Rain | $0.013 \pm 0.003$ | $0.014 \pm 0.002$ | $0.008 \pm 0.002$ | $0.383 \pm 0.089$ | $0.462 \pm 0.056$ | $0.175 \pm 0.011$ | $0.043 \pm 0.003$ |
| | NN5 | $0.090 \pm 0.011$ | $0.146 \pm 0.009$ | $0.603 \pm 0.107$ | $4.054 \pm 1.592$ | $1.372 \pm 0.180$ | $1.284 \pm 0.058$ | $0.473 \pm 0.135$ |
| | Fred-MD | $0.072 \pm 0.043$ | $0.118 \pm 0.051$ | $0.712 \pm 0.054$ | $5.371 \pm 1.455$ | $3.509 \pm 0.299$ | $0.376 \pm 0.025$ | $0.304 \pm 0.079$ |
| | Exchange | $0.240 \pm 0.112$ | $0.352 \pm 0.120$ | $1.984 \pm 0.836$ | $4.376 \pm 0.664$ | $1.583 \pm 0.932$ | $2.011 \pm 0.433$ | $0.455 \pm 0.268$ |

As shown in Table 3, BRIDGE consistently outperforms existing baselines across a variety of datasets. In terms of MDD, BRIDGE (w/o Text) ranks best on all but three datasets (i.e. pedestrian, where it ranks second and rain, traffic). For instance, on the Electricity dataset, BRIDGE attains an MDD of $0.206$, substantially lower than the second-best score model. Similarly, for the Wind dataset, BRIDGE's MDD of $0.365$ significantly outperforms the second-best score of $0.435$. The KL divergence results further underscore BRIDGE's capabilities, as it achieves the lowest K-L divergence on all the dataset, where only ranking second on pedestrian and rain dataset. Notably, for the Electricity dataset, BRIDGE's K-L divergence of $0.006$ is markedly better than the $0.008$ achieved by BRIDGE without text conditioning, and far superior to other models like TimeVQVAE ($0.203$) and TimeGAN ($0.507$) Interestingly, BRIDGE without text conditioning often achieves the second-best performance, suggesting that the core architecture of BRIDGE is robust even without additional textual information. For example, on the Pedestrian dataset, BRIDGE without text yields the best K-L divergence of $0.040$, closely followed by TimeVAE at $0.052$. It is worth noting that the proposed method significantly outperforms the DDPM. This indicates that, regardless of whether text is provided as additional input information, the proposed prototype mechanism can provide cross-domain contextual information to assist in generating target domain data. Furthermore, textual information in the form of word embeddings enhances this contextual information ( BRIDGE $vs.$ BRIDGE (w/o Text)), enabling the generation of more accurate target domain data. We also explored the impact of pre-training knowledge from LLMs. The results show that the larger models have a slight change in performance, but it is not significant, indicating that the pre-training knowledge has a minor influence on performance. Detailed results can be found in the Appendix H.

Table 4 shows the quality of generated data for the purposes of training models for downstream tasks. We generated synthetic data on two additional datasets to assist existing SOTA models in TS forecasting. All models were trained either using only real data or synthetic data and then tested on real test sets. The results indicate that training with only synthetic data can achieve comparable performance to real data across all models, as performance differences between real and synthetic data are less visible than differences in performance between architectures. This suggests that the generated data is sufficiently realistic, potentially allowing to share synthesised surrogates of otherwise sensitive data. For comparison, we also employed KernelSynth (Ansari et al., 2024) methods. Both methods effectively provided valuable synthetic data (compared to completely random data), but our proposed approach produced data that more closely resembles real data. This underscores its potential for generating meaningful synthetic data across domains.

Table 4: Comparison of MSE and MAE across various methods on time series forecasting. The results are for four different forecasting horizons: $H \in \{24, 36, 48, 60\}$ for ILI and $H \in \{6, 48\}$ for M4. Average results are reported. Full details in Appendix I.

| Dataset | | Random | | | LLM4TS | | | TEMPO | | | Time-LLM | | | GPT4TS | |
|---|---|---|---|---|---|---|---|---|---|---|---|---|---|---|---|
| | | MSE | MAE | | MSE | MAE | | MSE | MAE | | MSE | MAE | | MSE | MAE |
| ILI | Synthetic | | | | 1.98 | 0.89 | | 1.21 | 1.02 | | 2.20 | 1.44 | | 2.19 | 1.02 |
| | Real | 8.12 | 2.14 | | 1.86 | 0.86 | | 0.96 | 0.82 | | 2.00 | 1.20 | | 1.90 | 0.90 |
| | KernelSynth | | | | 4.35 | 1.50 | | 1.64 | 1.07 | | – | – | | 3.80 | 1.42 |

| Dataset | SMAPE | MASE | OWA | SMAPE | MASE | OWA | SMAPE | MASE | OWA | SMAPE | MASE | OWA | SMAPE | MASE | OWA |
|---|---|---|---|---|---|---|---|---|---|---|---|---|---|---|---|
| M4 Synthetic | | | | 12.82 | 1.92 | 0.97 | 12.10 | 1.66 | 0.88 | 12.78 | 3.06 | 1.24 | 12.82 | 1.91 | 0.97 |
| M4 Real | 24.603 | 3.895 | 1.925 | 12.08 | 1.67 | 0.89 | 11.88 | 1.61 | 0.86 | 12.33 | 2.87 | 0.89 | 12.36 | 1.77 | 0.92 |
| M4 KernelSynth | | | | 13.95 | 1.92 | 1.02 | 12.30 | 1.68 | 0.89 | - | - | - | 14.12 | 1.92 | 1.02 |

In order to verify that the semantic prototypes aid generalisation in the proposed model, we conducted few-shot learning on an unseen stock dataset. The models used were all trained on the mixed dataset. Table 5 shows that our model demonstrates robust few-shot capability, obtaining the best general MDD and K-L scores compared to the baselines. Additionally, more examples can further improve performance. This indicates that the proposed model can recall more accurate domain and pattern information from the learned semantic prototypes to assist in TSG.

| Methods | MDD | | K-L | |
|---|---|---|---|---|
| | 5-shots | 10-shots | 5-shots | 10-shots |
| TimeVQVAE | 3.502 | 3.514 | 2.311 | 4.685 |
| TimeGAN | 3.834 | 3.765 | 14.347 | 13.823 |
| GT-GAN | 3.653 | 3.474 | 10.971 | 8.855 |
| TimeVAE | 3.738 | 3.338 | 6.048 | 4.479 |
| Bridge | 3.421 | 3.107 | 2.349 | 2.827 |

Table 5: Few-shot Performance of Unseen Stock dataset. We compare the proposed methods and baseline on 5,10-shots. Best results are highlighted in bold face.

| Prototypes | 4 | 8 | 16 | 4 | 8 | 16 |
|---|---|---|---|---|---|---|
| Electricity | 0.460 | 0.232 | 0.173 | 0.006 | 0.005 | 0.005 |
| Solar | 397.136 | 378.011 | 375.530 | 0.102 | 0.042 | 0.034 |
| Wind | 0.655 | 0.387 | 0.347 | 0.075 | 0.099 | 0.086 |
| Traffic | 1.343 | 1.203 | 1.167 | 0.034 | 0.031 | 0.020 |
| Taxi | 0.848 | 0.647 | 0.588 | 0.104 | 0.072 | 0.069 |
| Pedestrian | 1.548 | 1.311 | 1.238 | 0.088 | 0.072 | 0.067 |
| Air | 0.879 | 0.742 | 0.637 | 0.039 | 0.034 | 0.028 |
| Temperature | 0.714 | 0.583 | 0.550 | 0.949 | 0.907 | 0.891 |
| Rain | 10.737 | 10.001 | 9.516 | 0.026 | 0.014 | 0.010 |
| NN5 | 1.950 | 1.432 | 1.352 | 0.288 | 0.146 | 0.088 |
| Fred-MD | 0.273 | 0.254 | 0.387 | 0.022 | 0.018 | 0.030 |
| Exchange | 0.416 | 0.398 | 0.394 | 0.141 | 0.112 | 0.132 |

(Left columns: Marginal Distribution Distance; Right columns: K-L Divergence)

Table 6: Ablation experiment on the impact of the number of prototypes. We experiment with the number of 4, 8, 16 separately.

## 6.4 ABLATION EXPERIMENT ON THE IMPACT OF PROTOTYPES AND TEXT

We further conducted ablation experiments. A s shown in Table 6, the number of prototypes significantly improves performance, indicating that the more prototypes there are, the more information they contain, which greatly aids the generation process. A representative generated sample can be seen in Figure 3. In general, conditional generation can significantly improve the accuracy and trend of numerical distributions. As shown in subfigure (2), without conditional control, the range of generated time series data is twice that of normal, while under conditions it is similar to the input. The performance of subfigure (1) is exactly the opposite. The value range of unconditional generation is greatly reduced in the final stage, and the gap with the input is obvious. In addition, unconditional generation also shows flaws in the trend in subgraph (2), and its fluctuation amplitude becomes significantly larger after 150 steps.

Figure 4 shows 16 semantic prototypes used in our text-to-time series generation model. Each prototype represents a distinct pattern in time series data, enabling the generation of diverse, domain-specific series. For example, prototypes {0,2,5} capture cyclical patterns useful for seasonal trends. Prototypes {6,7,13} represent trend patterns, including gradual changes and sharp transitions. Prototypes {1,3,4} show high-frequency fluctuations, representing volatility. By combining these pro-

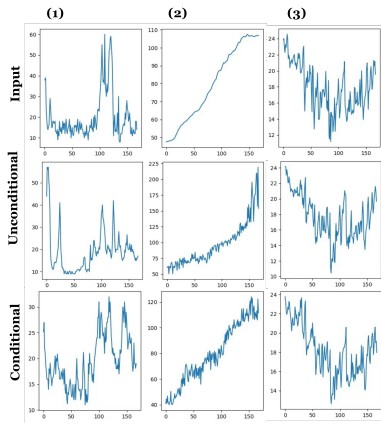

Figure 3: Visualisation of generation results from two different settings, where 'input' refers to the input data, 'unconditional' represents the result without text conditions, and 'conditional' refers to the generated result with text conditions.

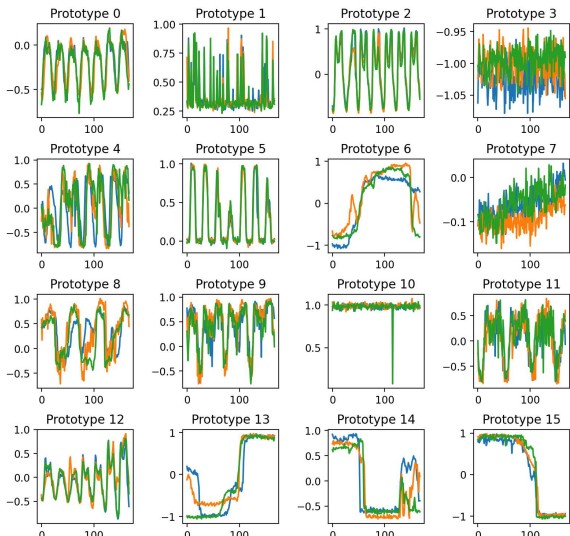

Figure 4: Visualization of semantic prototypes. Each prototype represents a different pattern or characteristic commonly found in time series data.

totypes, the model can generate rich, domain-specific time series data through translate text into time series data with specific semantic concepts. Figure 5 shows the distribution of prototypes across various domains. Some prototypes, like Prototype 3 in "kddcup" and "electricity," are widely relevant, while others, like Prototype 13 in "traffic," are domain-specific. The sparsity of the heatmaps shows that not all prototypes are equally important within a domain. For example, "rain" primarily uses prototypes {3,4,8}. This demonstrates the flexibility of the prototype-based approach, capturing both general and domain-specific patterns. Example generated data is shown in Appendix J.

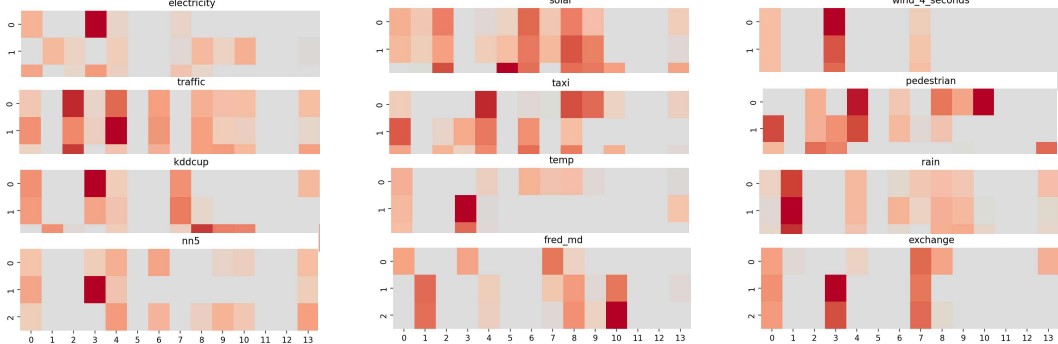

Figure 5: Prototype distribution across domains: Each heatmap shows prototype indices (x-axis, 0–15) and their frequency or importance (color intensity) in a specific domain

## 7 CONCLUSION

In this work, we explored the potential of using text to guide time series generation (TSG). We proposed a multi-agent system for optimizing time series textual descriptions, as well as a TSG model that incorporates text. Experiments demonstrate that concise text enhances TSG performance, with our model outperforming baselines, particularly in few-shot learning, thereby demonstrating strong generalization capabilities. Additionally, the results show that the designed semantic prototypes effectively utilize domain information. Our findings lay the groundwork for further advancing fully human-preferred text-based generation while also highlighting the challenges of this task.

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

## A   TEXT PREPARATION

### A.1   USING LLM DIRECTLY FOR TIME SERIES FORECASTING

Directly employed In-context learning (ICL) to activate LLMs for text generation is also considered. In this setup, the time series first adopts Seasonal-Trend decomposition using Loess (STL) (Cleveland et al., 1990), which is a robust method to decompose time series into long-term trend, seasonal, and residual components. Then, descriptions are generated separately for the initial, intermediate, final, and overall trends. It is important to note that this textual description is based on periodicity rather than time, as the time series is more nuanced. Descriptions segmented by time showed erroneous outputs in experiments, particularly in the form of regular fluctuations within specific intervals. For detailed prompt design consult. Figure 6 shown a example of using GPT4o directly generate time series with text and initial time series, result in stable fluctuation in a narrow range.

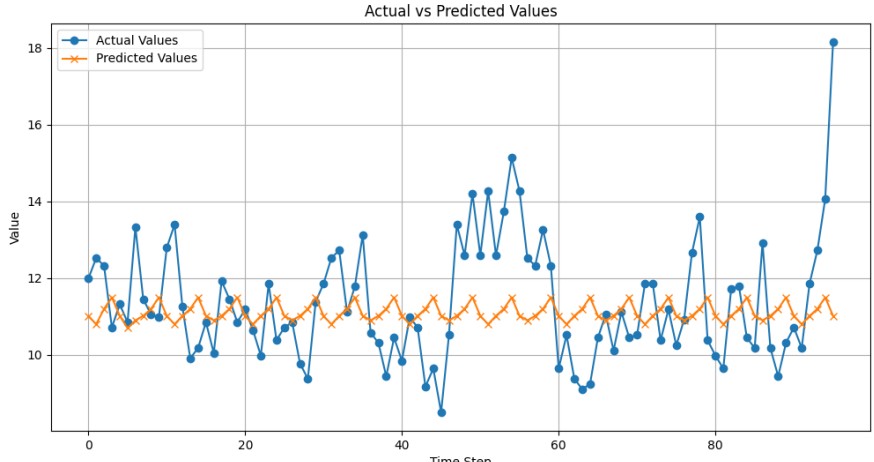

Figure 6: GPT4 directly output with text and time series as input.

### A.2   PIPELINE FOR COLLECT THE TEXT CANDIDATE

Figure 7 shows how the single agent framework is proposed how to collect templates and build an initial text description.

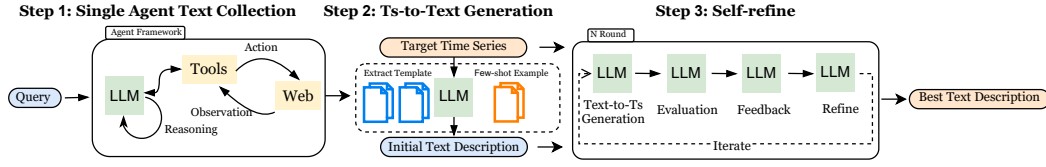

Figure 7: The pipeline of building text to time series dataset. The propsoed framework including three steps: *(i)* Leverage the ReAct to inspire agent collect human-craft text about time series description. *(ii)* Generate text description from given target time series dataset. *(iii)* iteratively refine the text description to fit the target time series.

### A.3   TEMPLATE BANK EXAMPLE

The time series templates extracted from the collected corpus typically contain descriptions of key patterns such as trends, seasonalities, and changes over time. For example, a typical template could be structured as:

*"Overall, {entity} {describe_general_trend}. At the beginning, {detail_initial}. As time progressed, {change_description}, culminating in {end_description} by {end_time}"*.

Additionally, the templates may include other relevant information, such as statistical metrics (e.g., minimum, maximum, standard deviation), dataset information, degree words (e.g., dramatically, slightly) that describe the intensity of changes, and the time series length."

## A.4 EVALUATION DIMENSIONS

In this section, we detail the initial evaluation criteria and definitions used to assess the generated text and its impact on TS forecasting. These criteria are designed to align with the modal characteristics of both text and time series, enabling the agent to evaluate and correct the input-output pairs effectively. We consider both text and TS metrics. Specifically, we consider following dimensions for text:

- **Accuracy of trend description:** The description accurately identifies the steady increase in the time series.
- **Mention of seasonality:** The description correctly notes the absence of seasonality in the data.
- **Completeness of information:** The description covers the main aspects of the time series but could mention the exact rate of increase.
- **Clarity of description:** The description is clear and easy to understand.

we consider following for time series: Specifically, we consider Mean Squared Error (MSE) (Hurvich, 1988), Kolmogorov-Smirnov Test (K-S Test) (Berger & Zhou, 2014) and Wasserstein Distance (WD) (Panaretos & Zemel, 2019) for measuring the difference between the generated and target time series, and building a 5-point Likert scale for evaluate the text quality with 5 dimension (i.e. Accuracy of trend description; Mention of seasonality; Reference to external factors; Clarity of description; Completeness of information).

## A.5 MULTI-AGENT COLLABORATION FRAMEWORK DETAILS

### A.5.1 FRAMEWORK WORKFLOW

We propose a structured, multi-agent collaboration framework designed to iteratively optimize text generation through systematic refinement. While the system is capable of operating with a single team employing distinct strategies, our experimental results demonstrate that employing two independent teams yields superior outcomes in terms of both quality and diversity of generated outputs. As can be seen from Figure 8, the framework comprises three primary stages:

In Stage 1: Task Planning, a manager agent assumes responsibility for overseeing the workflow. This agent coordinates all subsequent activities by distributing tasks and results from prior iterations to ensure seamless progress and alignment among team members. The manager also defines the objectives for the teams, thereby establishing a structured foundation for collaboration. Stage 2: Intra-group Collaboration constitutes the core of the system, wherein two independent teams of agents work concurrently to refine the given text. Each team is composed of four roles: a planner, a scientist, an engineer, and an observer. The planner serves as the team leader, formulating strategies and supervising operations. The scientist analyzes the input data and formulates detailed optimization plans. The engineer executes these plans, generating improved text outputs. The observer critically evaluates the plans and outputs, raising questions to identify shortcomings and potential improvements. Teams operate in iterative cycles, guided by the observer's critiques. This self-refining loop continues until the observer ceases to raise objections or a predefined maximum number of iterations is reached. Through this iterative process, each team independently produces a refined output. In Stage 3: Inter-group Discussion, the leaders of the two teams engage in a structured dialogue moderated by the manager. This stage facilitates the integration of insights from both teams, encouraging comparative evaluation and collaborative refinement of their outputs. The discussion continues until a consensus is reached, resulting in a unified solution that incorporates the strengths of both teams.

The finalized output is then subjected to Post-Processing. This phase includes a validation step, where the text is evaluated against a predefined model to ensure its quality and adherence to target

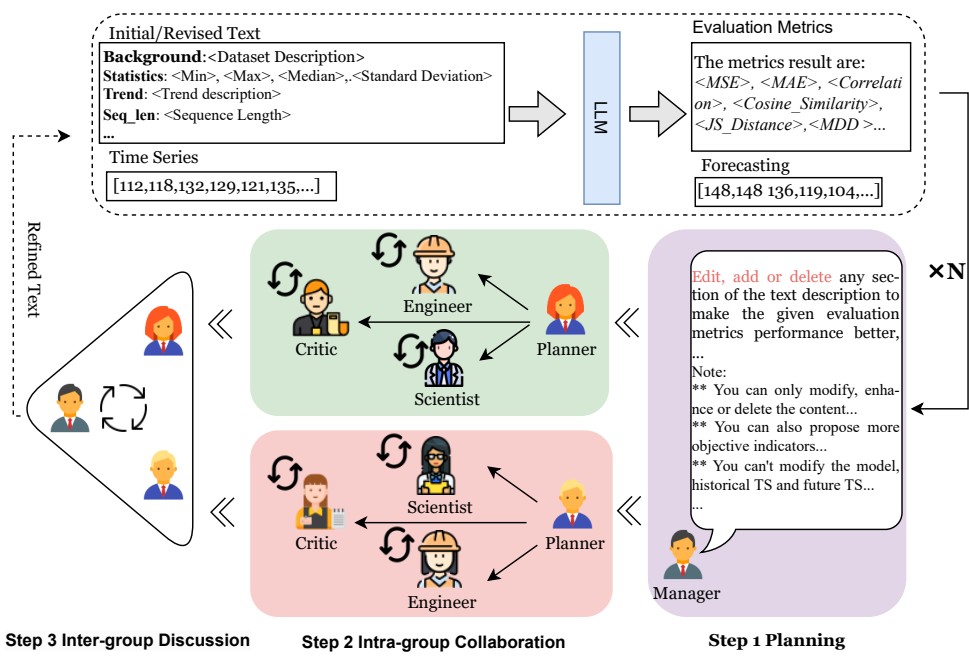

Figure 8: Detail workflow of proposed multi-agent collaborative framework

metrics. Approved outputs are incorporated into a formal dataset, expanding the training resources available for future tasks. Additionally, any templates developed during the process are added to a general template library, enabling reusability and continuous improvement in subsequent data generation efforts.

### A.5.2 ROLES AND RESPONSIBILITIES IN MULTI-AGENT SYSTEM

*Manager

- Lead and monitor the entire workflow of the system.
- Distribute tasks, data, and results from previous iterations to team leaders.
- Oversee inter-group discussions and ensure a consensus is reached.
- Approve final outputs and integrate them into the evaluation model for further refinement.

*Team Leaders (Planners)

- Plan and oversee the operations of their respective teams.
- Coordinate between team members to ensure tasks are completed efficiently.
- Represent their teams during inter-group discussions with the manager.
- Consolidate team outputs into a coherent proposal for refinement.

*Scientist

- Analyze the provided content and results.
- Formulate optimization plans to improve the text or dataset.
- Incorporate feedback from other team members, especially the observer, to refine strategies.
- Ensure that outputs align with optimization objectives.

*Engineer

- Implement the plans formulated by the scientist.
- Generate new text or refine existing content according to the plan.
- Provide iterative updates on progress to the team leader and scientist.
- Ensure that outputs meet the specified quality standards.

*Observer

- Critique the scientist's optimization plan, pointing out shortcomings or potential improvements.
- Question decisions to ensure robustness and completeness of the solution.
- Act as a quality control mechanism within the team, promoting thorough analysis.
- Signal the end of intra-group iterations when no further issues are identified.

### A.5.3 STAGES OF COLLABORATION

**Stage 1: Task Planning**
**Role Focus**: Manager

- The manager initiates the workflow and assigns responsibilities to the team leaders.

**Stage 2: Intra-group Collaboration**
**Role Focus**: Planner, Scientist, Engineer, and Observer

- Teams execute their tasks, with internal cycles involving:
    - The scientist proposing plans.
    - The engineer implementing them.
    - The observer providing feedback until quality is satisfactory.

**Stage 3: Inter-group Discussion**
**Role Focus**: Manager and Team Leaders

- Leaders from each team present their refined outputs to the manager.
- Discussions between leaders and the manager lead to a unified, optimized output.

### A.6 EXAMPLE OF INITIAL AND REFINED TEXT

**Initial Text:** This task focuses on predicting future values of a given time series based on historical data. The historical data shows periodic spikes in values, followed by declines, indicating a strong seasonal pattern. Notable fluctuations are observed at time points when values exceed 400, suggesting external influences. The average value of the historical data is approximately 239.95, with a variance of 8271.86, indicating significant fluctuations around the mean. Future Data Projections indicate that expected values for the time series range between a minimum of 310 and a maximum of 622, with a mean of 0.49 and a standard deviation of 92.75, suggesting increased variability compared to the historical period. The median value is projected to be around 453.34. Anticipated peaks are likely to occur at steps 5, 15, 25, and so on, while predicted dips are expected at steps 8, 18, 28, and similar intervals. Statistical Summary for the future time series reveals the following quartiles: the first quartile (Q1) is **, the median is **, and the third quartile (Q3) is **. Periodicity and Pattern Alignment show consistent high values expected every 10 steps, with auto-correlation anticipated around these intervals, reflecting the cyclical nature observed in the historical data.

**Iteration 1:** This dataset provides monthly totals of US airline passengers from 1949 to 1960. The prediction length is 29 time steps. **Future Data Description:** Future data projections indicate that expected values for the time series range between a **minimum of 310** and a **maximum of 622**, with a **mean of approximately 440.31** and a **standard deviation of about 79.52**, suggesting variability similar to the historical period. The **median value** is projected to be around **419**. The **first quartile (Q1)** is approximately **390.5**, and the **third quartile

---

**Algorithm 1** Generate Time Series from Text Input and Time Series Corpus

---

**Require:** K time series prompts $x$, prototype embeddings $P$
**Ensure:** Generated time series samples $\hat{x}$
 1: Initialise prototype embeddings $P$
 2: Extract prototype prompts $m$ with $D, T$
 3: Randomly sample noise $x_N \sim \mathcal{N}(0, I)$
 4: **for** $n$ from $N$ to 1 **do**
 5:    Predict step noise with $\tilde{\epsilon}_n = \tilde{\epsilon}_\theta(x_n, n, P, m)$
 6:    Denoise $x_{n-1} = \frac{x_n - \sqrt{1-\bar{\alpha}_n}\epsilon}{\sqrt{\bar{\alpha}_n}}$
 7: **end for**

---

(Q3)** is approximately **488.5**. Anticipated **peaks** are likely to occur at **steps 5, 15, and 25**, while predicted **dips** are expected at **steps 8, 18, and 28**, reflecting the cyclical patterns observed in the historical data. **Periodicity and pattern alignment** indicate consistent high values expected every **10 steps**, with **auto-correlation** anticipated around these intervals, continuing the cyclical nature observed previously.

**Iteration 2:** Future Data Projections indicate that expected values for the time series range between a minimum of 0.004 and a maximum of 0.98, with a mean of 0.49 and a standard deviation of 0.29, suggesting increased variability compared to the historical period. The median value is projected to be around 0.45. Anticipated peaks are likely to occur at steps 5, 15, 25, and so on, while predicted dips are expected at steps 8, 18, 28, and similar intervals. Statistical Summary for the future time series reveals the following quartiles: the first quartile (Q1) is 0.21, the median is 0.45, and the third quartile (Q3) is 0.70. Periodicity and Pattern Alignment show consistent high values expected every 10 steps, with auto-correlation anticipated around these intervals, reflecting the cyclical nature observed in the historical data.

**Iteration 3:** The air passengers dataset provides monthly totals of US airline passengers from 1949 to 1960. Data Projections indicate that expected values for the time series range between a minimum of 310 and a maximum of 622, with a mean of 453.34 and a standard deviation of 92.75, suggesting increased variability compared to the historical period. Anticipated peaks are likely to occur at steps 5, 15, 25, and so on, while predicted dips are expected at steps 8, 18, 28, and similar intervals.

**Refined Text:** The air passengers dataset provides monthly totals of US airline passengers from 1949 to 1960. The prediction length is 29 time steps. Data Projections indicate that expected values for the time series range between a minimum of 310 and a maximum of 622, with a mean of 453.34 and a standard deviation of 92.75, suggesting increased variability compared to the historical period. Anticipated peaks are likely to occur at steps 5, 15, 25, and so on, while predicted dips are expected at steps 8, 18, 28, and similar intervals.

### A.7 EXAMPLE RESULTS OF DIFFERENT STRATEGIES IN THE PROPOSED MULTI-AGENT SYSTEM.

Table 7,Table 8, Table 9 and Table 10 shows how multi agent system working together and decide what kind of text should be revised and provided. The agents take turns speaking, and ultimately the manager makes the decision.

## B DIFFUSION-BASED TIME SERIES GENERATION

### B.1 ALGORITHM FOR GENERATION

Algorithm 1 shown the diffusion process of generate new sample with text guide.

### B.2 THE FORM OF INPUT AND OUTPUT

Input of Diffusion Model including a time series and corresponding text description. The two of them will be processed by the encoder and LLama respectively, and the obtained embedding will

---

**Example Output of Collaboration Between Two Teams**

---

**Manager**

Thank you both for your detailed presentations. Let's break down the solutions from Team A and Team B to decide on the best overall solution.

Team A's Solution:

- Context and Scope: Provides a comprehensive introduction to the purpose and context.

- Detailed Observations: Adds contextual insights on historical data and anomalies.

- Statistical Summaries: Incorporates key statistical measures.

- Domain-Specific Insights: Utilizes domain knowledge for depth.

- Proposed Framework: Implements an iterative feedback loop.

Team B's Solution:

- Clarity and Specificity: Emphasizes enhancing understanding through clarity and statistical depth.

- In-depth Statistical Summaries: Provides detailed statistical measures, including quartiles and auto-correlation.

- Periodic Trends: Highlights specific expected patterns and trends.

- Example Descriptions: Well-structured and detailed historical and future descriptions with statistical insights.

- Overall Metrics and Patterns: Integrated cleanly with example descriptions, improving the predictive model's depth.

Key Comparisons and Considerations:

- Introduction and Context:
    - Team A adds more contextual background to enhance understanding of the task.
    - Team B focuses on the specifics of the data without much introductory context.

- Statistical Depth:
    - Team B goes deeper into statistical measures, including quartiles and auto-correlation, which may provide a more comprehensive understanding.
    - Team A offers core statistical insights but may lack some of the depth seen in Team B's approach.

- Clarity and Readability:
    - Both teams provide clear and readable descriptions, but Team B's focus on eliminating redundancy improves conciseness.

---

Table 7: The demo output of the multi-agent collaboration framework for iterative text revision.

---

**Decision**

---

**Team A:**
Introduces the task well, provides core statistical measures, includes domain-specific insights, and proposes a valuable iterative improvement framework.

**Team B:**
Offers more detailed statistical summaries, clear patterns, and a reduction of redundant information, improving clarity and potentially enhancing the predictive model's accuracy.

**Final Decision:**
Incorporating the strengths of both solutions would be ideal. However, if we must choose one, **Team B's solution** edges slightly ahead due to its detailed statistical depth and the refined approach to clarifying and specifying trends and patterns. This comprehensive statistical summary can significantly enhance the predictive model's accuracy, which is critical for improving predictions.

Does everyone agree with this decision, or are there any additional factors or preferences that should be considered before finalizing our choice?

---

Table 8: Summary of the decision-making process

---

**Team_A_Planner and Team_B_Planner Discuss**

---

**Team A_Planner:**
Thank you for the thorough comparison. I agree that incorporating elements from both Team A and Team B would yield the most robust solution. However, since we must choose one, Team B's solution appears to offer a deeper statistical analysis which can be extremely beneficial for improving the predictive model's accuracy.

**Decision Recap:**

- **Context and Clarity:**
  - Team A's contextual introduction adds understanding but Team B's specific focus on the task at hand makes their text more to the point.

- **Statistical Depth:**
  - Team B's detailed statistical summaries, including quartiles and auto-correlation, provide a richer dataset for analysis.

- **Readability and Redundancy:**
  - Team B's elimination of redundancy and clear structure enhances readability and makes the information more accessible.

- **Iterative Feedback:**
  - Not explicitly addressed here but likely discussed in greater detail within Team A's framework.

**Team_B_Planner:**
I appreciate the detailed analysis and the balanced approach to deciding between the two solutions. Integrating elements from both teams leverages their strengths to create a more robust and comprehensive plan.

---

Table 9: Evaluation of Team A and Team B's solutions.

**Final Implementation Plan Analysis**

**Consensus Plan:**
**Adoption of Team B's Detailed Statistical Approach:**

- Use Team B's detailed statistical summaries, including quartiles and auto-correlation, to provide a comprehensive understanding.
- Emphasize clarity and specificity to enhance the readability and usefulness of the descriptions.

**Incorporation of Team A's Context, Iterative Feedback, and Domain Insights:**

- Add a brief introduction from Team A's plan to set the context and purpose.
- Implement Team A's iterative feedback approach for continuous refinement based on prediction performance.
- Wherever applicable, integrate domain-specific insights to provide additional depth and relevance.

**Final Implementation Plan List:**
**Historical Data Analysis:**
The historical data analysis focuses on identifying key patterns, trends, and statistical insights.

**Periods of High Volatility:**
Time steps 1–20 show values fluctuating between [Min: 0.03, Max: 0.99, Avg: 0.55].
**Stable Periods:**
Time steps 21–40 exhibit a narrower range with values [Min: 0.18, Max: 0.76, Avg: 0.45].
**Detailed Trends:**

- Significant peaks at time steps: 1, 4, 8, 12, 16.
- Consistent dips at time steps: 30, 50, 70.

**Statistical Insights:**

- **Overall Metrics:** Min: 0.001, Max: 0.996, Mean: 0.50, Std Dev: 0.26.
- **Quartile Ranges:** Q1: Min: 0.01, Max: 0.79, Mean: 0.45.

Table 10: Summary of the Consensus Plan and Final Implementation Steps.

be fused through a single-layer MLP as conditional input. The output of the diffusion model is a synthetic time series.

## B.3 MODEL ARCHITECTURE

We provide the outline of proposed diffusion model architecture for a single UNet block

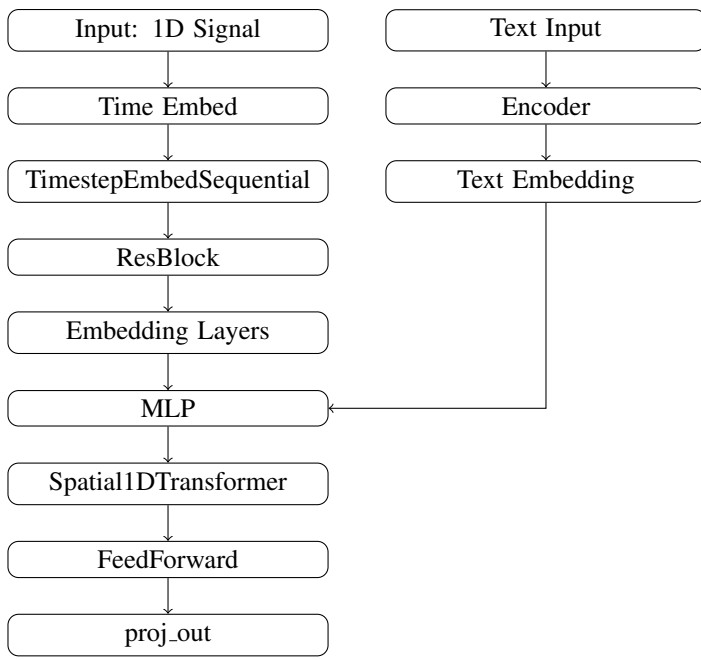

## C BASELINE MODEL

### C.1 TIME SERIES GENERATION MODEL

**TimeVQVAE**(Lee et al., 2023) is a generative model designed for sequential data. It combines the strengths of a variational autoencoder (VAE) with vector quantisation to discretise latent space representations, making it effective for time series data. The model consists of an encoder that compresses the input data into a discrete latent space and a decoder that reconstructs the time series. TimeVQVAE is particularly useful for generating realistic time series samples while maintaining key temporal dependencies. The quantisation step helps in learning discrete representations that can be reused for efficient time series modelling and generation.

**TimegGAN** (Yoon et al., 2019) is a variant of the GAN framework specifically tailored for time series data. It combines both supervised and unsupervised learning approaches, using a generator to create synthetic time series and a discriminator to differentiate between real and generated data. Additionally, it integrates an embedding network to capture temporal dependencies and preserve temporal correlations between generated samples. The model ensures that the generated time series not only closely mimic the statistical properties of the original data but also maintain the correct temporal ordering and dynamics. TimegGAN is particularly useful in applications requiring realistic synthetic data generation, such as forecasting and anomaly detection.

**GT-GAN** (Jeon et al., 2022) introduces a novel architecture for time series generation by incorporating both global and local perspectives. The model features two generators: one focuses on capturing the global trends across the entire time series, while the other focuses on local variations. The two components work together to ensure that the generated time series exhibit realistic patterns on both macro and micro levels. GT-GAN uses a two-stream discriminator that evaluates both the global and local outputs, ensuring high fidelity in the generated data. This model is effective for generating complex time series where both long-term trends and short-term fluctuations are important.

**TimeVAE** (Desai et al., 2021) extends the traditional VAE architecture to model time series data. It uses an encoder to map time series data into a continuous latent space, from which the decoder reconstructs the original time series. The model captures uncertainty and variation in the data through the latent space's probabilistic structure, making it well-suited for applications where capturing latent factors and generating multiple plausible future scenarios is important. TimeVAE can be applied to various tasks, such as anomaly detection, forecasting, and data augmentation, by learning complex temporal dependencies and generating realistic time series that adhere to the original data's statistical properties.

**Denoising Diffusion Probabilistic Models (DDPM)** (Ho et al., 2020) extend the traditional generative model framework by utilizing a diffusion process to model data generation. DDPMs begin with a Gaussian noise and iteratively refine it through a reverse diffusion process, gradually transforming the noise into a realistic data sample. This process involves a series of denoising steps where the model learns to remove noise from the data at each step, allowing for high-quality data generation.

## C.2 TIME SERIES FORECASTING MODEL

**Time-LLM** (Jin et al., 2023) is a powerful TS LLM that outperforms specialized forecasting models, which repurposes LLMs for time series forecasting by reprogramming input data and employing the Prompt-as-Prefix (PaP) technique for enhanced context alignment.

**GPT4TS** (Zhou et al., 2023a) takes advantage of pre-trained language and vision models for general time series analysis. By demonstrating that supervised fine-tuning (SFT) can successfully extend LLM capabilities to time series tasks, GPT4TS bridges the gap between natural language processing models and temporal data analysis. The model's architecture shows the feasibility of applying large pre-trained models to time series, leading to significant performance improvements in various time series applications.

**LLM4TS** (Chang et al., 2023) is an innovative framework that repurposes pre-trained LLMs for time-series forecasting, employing a two-stage fine-tuning strategy and a two-level aggregation method to align with and enhance the model's ability to process multi-scale temporal data, outperforming state-of-the-art models in both fune-tuing and few-shot scenarios.

**TEMPO** (Cao et al., 2024) proposed using prompts to adapt to different time series distributions. It demonstrates superior performance in zero-shot settings across diverse benchmark datasets, showcasing its potential as a foundational model-building framework for capturing dynamic temporal phenomena.

## D EXPERIMENT SETUP

The time series length $T$ for generation is set to 168 in a form of non-overlap uni-variate sequence slices for all the datasets. For forecasting, we assessed performance over four different prediction horizons $H \in \{24, 36, 48, 60\}$ for ILI and $H \in \{6, 48\}$ for M4.

## E IMPLEMENTATION DETAIL

We implemented all the model and conduct all experiments on single NVIDIA Tesla H/A100 80GB GPUs. For LLM used in proposed model is LLama3-8B (Dubey et al., 2024). For generation task, we keep all model's sequence length is 168 which is the max length of `Pedestrian`, `Rain`, `Temperature` datasets. For evaluation of the synthesis data quality task, we keep the sequence length of 256.

The reported result are all under following training settings. The number of prototypes are set to 16 for all the main evaluations. Models for each sequence length are trained for 50, 000 steps using a batch size of 128 and a learning rate of $5 * 10^{-5}$ with 1, 000 warm-up steps.

## F  DATASET ANALYSIS

### F.1  DETAILS OF DATASETS

In this section, we provide a detailed overview of the datasets used for model training in this paper:

**Electricity:** This dataset captures hourly electricity consumption for 321 clients between 2012 and 2014, measured in kilowatts (kW). It was originally sourced from the UCI repository.

**Solar:** Comprising 137 time series, this dataset records hourly solar power production in the state of Alabama throughout 2006.

**Wind:** Wind: This dataset includes a single, extensive daily time series that tracks wind power production (in megawatts) at 4-second intervals, starting from August 1, 2019. It was obtained from the Australian Energy Market Operator (AEMO) platform.

**Traffic:** Covering 15 months of daily data (440 records), this dataset represents the occupancy rate (ranging from 0 to 1) of various car lanes on the San Francisco Bay Area freeways over time.

**Taxi** This dataset contains spatio-temporal traffic time series of New York City taxi rides, recorded every 30 minutes at 1,214 locations during January 2015 and January 2016.

**Pedestrian:** Featuring hourly pedestrian counts from 66 sensors in Melbourne, this dataset spans from May 2009 to April 30, 2020, and is regularly updated as new data becomes available.

**Air Quality:** Used in the KDD Cup 2018 forecasting competition, this dataset includes hourly air quality measurements from 59 stations in Beijing (35 stations) and London (24 stations) between January 1, 2017, and March 31, 2018. The data includes various air quality metrics such as PM2.5, PM10, NO2, CO, O3, and SO2. Missing values were imputed using leading zeros or the Last Observation Carried Forward (LOCF) method.

**Temperature:** This dataset consists of 32,072 daily time series with temperature observations and rain forecasts from 422 weather stations across Australia, collected between May 2, 2015, and April 26, 2017. Missing values were replaced with zeros, and the mean temperature column was extracted for use.

**Rain:** Similar to the Temperature dataset, this dataset focuses on rain data extracted from the same source.

**NN5:** Used in the NN5 forecasting competition, this dataset contains 111 time series from the banking sector, with the goal of predicting daily cash withdrawals from ATMs in the UK. Missing values were replaced by the median of the same weekday across the series.

**Fred-MD:** This dataset contains 107 monthly time series reflecting various macroeconomic indicators, sourced from the Federal Reserve Bank's FRED-MD database. The series have been differenced and log-transformed following established practices in the literature.

**Exchange:** This dataset records daily exchange rates for eight currencies.

**Stock:** This dataset consists of daily stock prices for the symbol GOOG, which is listed on NASDAQ.

### F.2  DATASET STATISTICS

To test the quality of the synthetic data generated by our proposed model, we conducted tests on two additional datasets. In the experiments, we trained the synthetic data to be the same as the original data and tested it on the real datasets. The statistics of the datasets are in Table 11:

## G  EVALUATION METRICS

The calculations of these metrics are as follows:

| Domain | Tasks | Datasets | Dim. | Series Length | Dataset Size | Frequency |
|---|---|---|---|---|---|---|
| Long-Term | ILI | 7 | 24, 36, 48, 60 | (617, 74, 170) | 1 week | Illness |
| Short-term Forecasting | M4-Yearly | 1 | 6 | (23000, 0, 23000) | Yearly | Demographic |
| | M4-Quarterly | 1 | 8 | (24000, 0, 24000) | Quarterly | Finance |
| | M4-Monthly | 1 | 18 | (48000, 0, 48000) | Monthly | Industry |
| | M4-Weekly | 1 | 13 | (359, 0, 359) | Weekly | Macro |
| | M4-Daily | 1 | 14 | (4227, 0, 4227) | Daily | Micro |
| | M4-Hourly | 1 | 48 | (414, 0, 414) | Hourly | Other |

Table 11: Comparison of datasets for long-term and short-term forecasting tasks

$$\text{MSE} = \frac{1}{H} \sum_{h=1}^{H} (Y_h - \hat{Y}_h)^2, \qquad \text{MAE} = \frac{1}{H} \sum_{h=1}^{H} |Y_h - \hat{Y}_h|,$$

$$\text{SMAPE} = \frac{200}{H} \sum_{h=1}^{H} \frac{|Y_h - \hat{Y}_h|}{|Y_h| + |\hat{Y}_h|}, \qquad \text{MAPE} = \frac{100}{H} \sum_{h=1}^{H} \frac{|Y_h - \hat{Y}_h|}{|Y_h|},$$

$$\text{MASE} = \frac{1}{H} \sum_{h=1}^{H} \frac{|Y_h - \hat{Y}_h|}{\frac{1}{H-s} \sum_{j=s+1}^{H} |Y_j - Y_{j-s}|}, \qquad \text{OWA} = \frac{1}{2} \left( \frac{\text{SMAPE}}{\text{SMAPE}_{\text{Naïve2}}} + \frac{\text{MASE}}{\text{MASE}_{\text{Naïve2}}} \right),$$

where $s$ is the periodicity of the time series data, $H$ denotes the number of data points (i.e., prediction horizon in our cases), and $Y_h$ and $\hat{Y}_h$ are the $h$-th ground truth and prediction, where $h \in \{1, \ldots, H\}$.

For generation, we consider Marginal Distribution Difference (MDD):

$$\text{MDD}(P, Q) = \sum_{x \in X} |P(x) - Q(x)|$$

where $P$ and $Q$ represent the marginal distributions of the real and synthetic data, and $X$ denotes the set of possible values for the variable being analyzed.

Also Kullback-Leibler divergence (K-L)

$$D_{KL}(P \| Q) = \sum_{x \in X} P(x) \log \left( \frac{P(x)}{Q(x)} \right)$$

where $P$ and $Q$ are the two probability distributions being compared, and $X$ represents the set of possible values.

## H  THE IMPACT OF LLMs ON THE DIFFUSION MODEL PERFORMANCE

The Table 12 compares the performance of Llama and GPT2 as encoders in our diffusion model across various time series domains. Both models show similar performance in most domains, with slight differences in specific cases. For example, Llama performs slightly better in the "Electricity" (0.173 vs 0.208) and "Rain" (0.387 vs 0.427) domains, suggesting a better ability to capture fluctuations in these time series. In contrast, GPT2 outperforms Llama in "Air" (0.655 vs 0.637) and "Temperature" (0.612 vs 0.550), indicating its strength in encoding gradual trends. Overall, both models show strong performance across multiple domains, with only minor variations. These results highlight that while Llama and GPT2 differ slightly in their handling of specific time series patterns, both are effective encoders for our diffusion model, capable of capturing both domain-specific and general temporal features.

## I  DATA AUGMENTATION RESULTS

For long-term forecasting (Table 13), we find that the LLM4TS trained via the synthetic data produces relatively low MSE and MAE values, such as ILI-24 Synthesis with an MSE of 1.84 and an

| Model | Electricity | Solar | Wind | Traffic | Taxi | Pedestrian |
|---|---|---|---|---|---|---|
| Llama | 0.173 | 375.530 | 0.347 | 1.167 | 0.588 | 1.238 |
| GPT2 | 0.208 | 375.538 | 0.356 | 1.189 | 0.624 | 1.143 |
| | **Air** | **Temperature** | **Rain** | **NN5** | **Fred-MD** | **Exchange** |
| Llama | 0.637 | 0.550 | 9.516 | 1.352 | 0.387 | 0.394 |
| GPT2 | 0.655 | 0.612 | 9.833 | 1.351 | 0.427 | 0.458 |

Table 12: Model performance across different domains. Result measured by MDD

MAE of 0.85, which are competitive with the performance on real-world datasets. In fact, for length like 24 and 36, LLM4TS consistently performs well, showing competitive results in both MSE and MAE, even when compared to training on real data. GPT4TS and Time-LLM, on the other hand, exhibit a slight drop in performance when trained on synthetic data, but considerable accepted. In the short-term forecasting scenario (Table 14), the results show similar trends. For example, in the M4-Hourly Synthesis, LLM4TS achieves a competitive SMAPE of 33.06 and MASE of 10.252 when trained on synthetic data, closely matching its performance on real data. This suggests that synthetic data can effectively simulate real data patterns, making it a viable option for model training when real-world data is limited or unavailable.

| Methods | LLM4TS | | TEMPO | | Time-LLM | | GPT4TS | |
|---|---|---|---|---|---|---|---|---|
| Metrics | MSE | MAE | MSE | MAE | MSE | MAE | MSE | MAE |
| *ILI-24 KernelSynth* | 4.36 | 1.49 | 1.48 | 1.02 | - | - | 3.92 | 1.45 |
| *ILI-36 KernelSynth* | 4.32 | 1.49 | 1.37 | 0.96 | - | - | 3.87 | 1.43 |
| *ILI-48 KernelSynth* | 4.15 | 1.48 | 1.69 | 1.09 | - | - | 3.77 | 1.40 |
| *ILI-60 KernelSynth* | 4.35 | 1.50 | 2.01 | 1.22 | - | - | 3.62 | 1.39 |
| *ILI-24 Ours* | 1.84 | 0.85 | 1.00 | 0.87 | 2.05 | 1.29 | 2.23 | 0.99 |
| *ILI-36 Ours* | 1.86 | 0.86 | 1.22 | 0.99 | 2.13 | 1.34 | 2.13 | 0.97 |
| *ILI-48 Ours* | 1.88 | 0.88 | 1.34 | 1.08 | 2.35 | 1.60 | 2.28 | 1.05 |
| *ILI-60 Ours* | 2.37 | 0.99 | 1.49 | 1.14 | 2.30 | 1.55 | 2.35 | 1.09 |
| *ILI-24 Real* | 1.78 | 0.81 | 0.66 | 0.63 | 1.83 | 1.15 | 1.99 | 0.88 |
| *ILI-36 Real* | 1.75 | 0.82 | 0.92 | 0.80 | 1.90 | 1.17 | 1.90 | 0.90 |
| *ILI-48 Real* | 1.72 | 0.84 | 1.33 | 1.02 | 2.16 | 1.26 | 1.81 | 0.88 |
| *ILI-60 Real* | 2.20 | 0.95 | 0.91 | 0.80 | 2.11 | 1.23 | 1.87 | 0.92 |

Table 13: Comparison of MSE and MAE across various methods on Long-term forecasting. The results are for four different forecasting horizons: $H \in \{24, 36, 48, 60\}$. Red values indicate the best score, and blue values represent the second best.

## J PROTOTYPES SAMPLE RESULT

Figure 9 represents the corresponding data visualization of different domains.

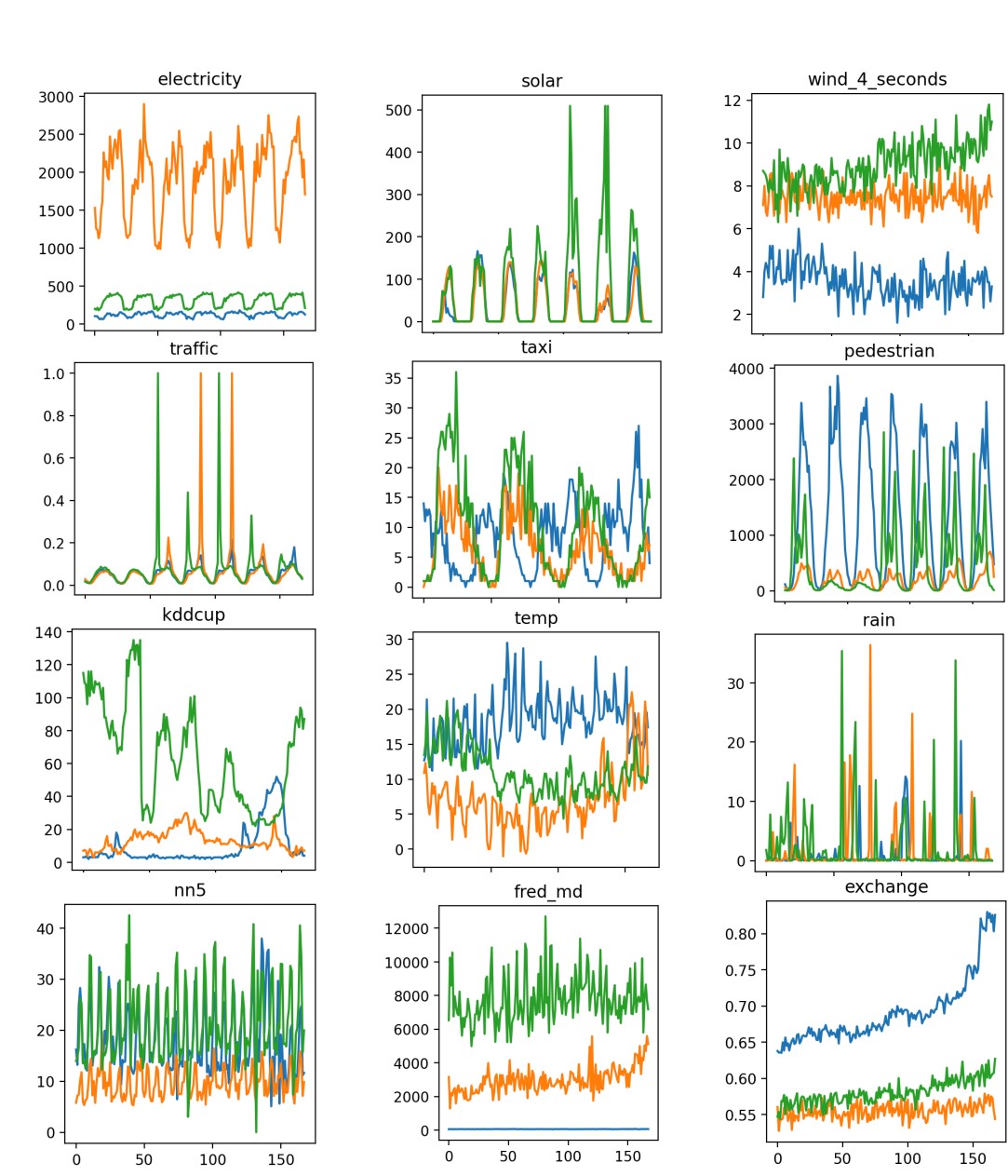

Figure 9: Visualize data in different domains

| Methods | Random | | | LLM4TS | | | TEMPO | | | Time-LLM | | | GPT4TS | | |
|---|---|---|---|---|---|---|---|---|---|---|---|---|---|---|---|
| | SMAPE | MASE | OWA | SMAPE | MASE | OWA | SMAPE | MASE | OWA | SMAPE | MASE | OWA | SMAPE | MASE | OWA |
| *M4-Hourly KernelSynth* | - | - | - | 21.662 | 3.415 | 1.302 | 19.768 | 1.583 | 0.868 | - | - | - | 21.339 | 2.252 | 1.050 |
| *M4-Daily KernelSynth* | - | - | - | 3.601 | 3.975 | 1.198 | 3.498 | 3.877 | 1.166 | - | - | - | 3.700 | 3.979 | 1.214 |
| *M4-Weekly KernelSynth* | - | - | - | 10.665 | 3.667 | 1.242 | 10.248 | 3.447 | 1.180 | - | - | - | 10.799 | 3.965 | 1.303 |
| *M4-Monthly KernelSynth* | - | - | - | 14.477 | 1.077 | 1.008 | 13.991 | 1.066 | 0.986 | - | - | - | 14.695 | 1.095 | 1.024 |
| *M4-Quarterly KernelSynth* | - | - | - | 12.063 | 1.457 | 1.079 | 11.784 | 1.422 | 1.053 | - | - | - | 11.971 | 1.414 | 1.059 |
| *M4-Yearly KernelSynth* | - | - | - | 16.619 | 3.743 | 0.979 | 16.051 | 3.513 | 0.933 | - | - | - | 17.008 | 3.733 | 0.990 |
| *Average* | - | - | - | 13.946 | 1.923 | 1.017 | 12.304 | 1.682 | 0.892 | - | - | - | 14.122 | 1.915 | 1.021 |
| *M4-Hourly Ours* | - | - | - | 33.06 | 10.252 | 3.039 | 25.942 | 7.532 | 2.278 | 22.435 | 4.899 | 1.726 | 33.06 | 10.252 | 3.039 |
| *M4-Daily Ours* | - | - | - | 4.749 | 5.391 | 1.602 | 3.606 | 3.997 | 1.202 | 3.891 | 4.012 | 1.411 | 4.749 | 5.391 | 1.602 |
| *M4-Weekly Ours* | - | - | - | 12.979 | 5.196 | 1.644 | 11.905 | 3.97 | 1.365 | 11.850 | 3.762 | 1.355 | 12.979 | 5.196 | 1.644 |
| *M4-Monthly Ours* | - | - | - | 13.157 | 0.981 | 0.917 | 12.975 | 0.96 | 0.901 | 13.877 | 1.111 | 1.017 | 13.157 | 0.981 | 0.917 |
| *M4-Quarterly Ours* | - | - | - | 10.608 | 1.253 | 0.939 | 10.318 | 1.207 | 0.909 | 10.877 | 1.342 | 1.022 | 10.608 | 1.253 | 0.939 |
| *M4-Yearly Ours* | - | - | - | 15.547 | 3.72 | 0.944 | 13.466 | 3.036 | 0.794 | 13.788 | 3.255 | 0.843 | 15.547 | 3.72 | 0.944 |
| *Average* | - | - | - | 12.821 | 1.916 | 0.974 | 12.104 | 1.663 | 0.881 | 12.786 | 3.063 | 1.235 | 12.821 | 1.916 | 0.974 |
| *M4-Hourly Real* | 49.163 | 16.089 | 4.696 | 18.356 | 2.972 | 1.120 | 22.847 | 5.323 | 1.733 | 20.323 | 4.573 | 1.507 | 20.642 | 4.070 | 1.411 |
| *M4-Daily Real* | 4.97 | 5.531 | 1.66 | 3.224 | 3.452 | 1.056 | 3.052 | 3.251 | 0.997 | 3.376 | 3.651 | 1.111 | 3.205 | 3.455 | 1.053 |
| *M4-Weekly Real* | 15.084 | 5.533 | 1.819 | 12.400 | 4.848 | 1.550 | 10.544 | 3.377 | 1.183 | 11.330 | 3.666 | 1.278 | 12.433 | 4.779 | 1.539 |
| *M4-Monthly Real* | 22.756 | 1.959 | 1.71 | 12.817 | 0.947 | 0.890 | 12.698 | 0.934 | 0.879 | 13.327 | 1.023 | 0.943 | 12.916 | 0.958 | 0.898 |
| *M4-Quarterly Real* | 19.216 | 2.587 | 1.816 | 10.301 | 1.207 | 0.908 | 10.077 | 1.177 | 0.887 | 10.672 | 1.266 | 0.946 | 10.386 | 1.230 | 0.920 |
| *M4-Yearly Real* | 37.396 | 8.755 | 2.246 | 13.885 | 3.240 | 0.833 | 13.493 | 3.052 | 0.797 | 13.498 | 3.013 | 0.792 | 14.801 | 3.633 | 0.910 |
| *Average* | 24.603 | 3.895 | 1.925 | 12.075 | 1.665 | 0.881 | 11.878 | 1.604 | 0.857 | 12.330 | 2.865 | 0.892 | 12.362 | 1.771 | 0.919 |

Table 14: Time series forecasting results on unseen time series dataset. The forecasting horizons are in [6, 48] and report value is the average. A lower value indicates better performance. Red: the best, Blue: the second best.

