# OpenReview forum: "BRIDGE: Bootstrapping Text to Guide Time-Series Generation via Multi-Agent Iterative Optimisation and Diffusion Modelling"
_ICLR.cc/2025/Conference — Submitted to ICLR 2025_

### Official Review · Reviewer_x5xf · 2024-10-29

**Soundness:** 3
**Presentation:** 4
**Contribution:** 3
**Rating:** 8
**Confidence:** 4

**Summary:**

The authors propose a new method for generating high quality time-series from text descriptions. They first propose a multi-agent setup for refining text descriptions of time-series. They also propose a new diffusion-based method for generating time-series data. They test which components of textual descriptions have the greatest impact on performance, the quality of the generated time-series (with and without text) and the performance of models trained on synthesized data compared to real data.

**Strengths:**

Overall, I think this is a strong paper and tackles a new and emerging issue in time-series analysis, which is building high quality datasets, particularly with text components.
1. They propose both a new time-series text refinement approach and a time-series generation method. I think having both of those components improves the strength of the paper dramatically.
2. They propose a multi-agent approach for text refinement which appears to significantly improve the quality of the text descriptions
3. Multi-agent dataset generation seems to be relatively understudied within the time-series domain.
4. Their diffusion based time-series approach generates high quality time-series data with diverse properties and is useful for training models.
5. They also test which components of the text descriptions are impacting performance which I thought was very interesting, and they test their generated time-series quality in terms of both pure generation quality and for model training purposes.

**Weaknesses:**

1. No code provided
2. No data exampled provided. Particularly I would like to see more examples of refined text paired with time-series. This would also make it easier to understand the overall quality of the text and time-series generation.
3. It is difficult to disentangle which parts of the agent architecture contribute to the improved text descriptions.

**Questions:**

1. Are you going to release the code for this project?
2. The authors highlight the usefulness of the data, particularly for combined text and time-series descriptions. Why not make and release a collection of these? It would also improve the impact fullness of this work.
3. How would removing parts of the agent architecture impact performance? What about different text models?

---

> ### Author Response · Authors · 2024-11-22
> **Kindly Reply**
>
> Thank you for your really kind advice. We really appreciate your suggestion and already revise them in the new version of papers.
>
> **❓Are you going to release the code and dataset for this project?**
>
> 💡 Thank you for your interest in our work. We plan to release the code and datasets used for the benchmark and all experiments in the paper, under an Apache-2.0 license upon acceptance. Currently,  Anonymized code is available here: [](https://anonymous.4open.science/r/BRIDGE-CB58/LICENSE)
>
> https://anonymous.4open.science/r/BRIDGE-CB58
>
> **❓How would removing parts of the agent architecture impact performance?**
>
> 💡 In the experiment, the two teams showed completely different strategies: one team cared more about details in text description, while the other cared more about the overall trend and statistics. Specifically, we compare the single team with multi teams. Macro refers to a single team executing high-level information adjustments, while Micro focuses on details. Multiple teams indicate collaboration between two teams to complete the task.   Overall, collaboration among multiple teams outperforms any single-team strategy, indicating that combining different strategies leads to more comprehensive and appropriate textual outputs. From a strategic perspective, the macro single-team approach performs better than the micro single-team approach, suggesting that overly detailed textual descriptions are still challenging to utilize effectively at this stage. Both teams chose to include statistical information, aligning with previous work that these factors most intuitively provide valuable insights, detail example can be find in Appendix A.5.
>
> | Policy | Airpassenger (LLMTime) | Airpassenger (LSTPrompt) | Sunspots (LLMTime) | Sunspots (LSTPrompt) |
> | --- | --- | --- | --- | --- |
> | Multi | 40.94 | 12.39 | 48.64 | 42.37 |
> | Single (Micro) | 44.27 | 14.22 | 56.80 | 45.70 |
> | Single (Macro) | 42.57 | 13.83 | 54.51 | 45.01 |
>
> **❓What about different text models?**
>
> 💡 Thank you for the reminder. We have conducted an ablation experiment on two different text models. For more details please see our response to reviewer ZPw4
>
> **❓It is difficult to disentangle which parts of the agent architecture contribute to the improved text descriptions.**
>
> 💡The proposed LLM-based multi-agent systems mainly has two components, one responsible for testing the performance of the text and the other responsible for refining the text, and they play equally important roles. The former is validated using a model that is widely used as a baseline and allows text as part of the input, where we hope that the results will be more reliable. The latter is responsible for modification, which includes an agent (manager) responsible for monitor the entire process and two teams responsible for analysis and modification respectively. We have rewritten this section in the section 3 and appendix A.5.
>
> **Comments, Suggestions And Typos:** Thank you for your advice. We address all of them in the revised version. Since we have answered all your questions and have added all your requested experiments, please consider raising the score to give us an opportunity to present our work at proceedings.

---

> ### Author Response · Authors · 2024-11-22
> **Kindly Reply 2**
>
> **❓ Can you provide more examples of refined text paired with time series?**
>
> 💡 We provide an example of how the text description evolved with more iterations iteration.
>
> | Iteration | Text |
> |----------|----------|
> | 1    | This task focuses on predicting future values of a given time series based on historical data. The historical data shows periodic spikes in values, followed by declines, indicating a strong seasonal pattern. Notable fluctuations are observed at time points when values exceed 400, suggesting external influences. The average value of the historical data is approximately 239.95, with a variance of 8271.86, indicating significant fluctuations around the mean. Future Data Projections indicate that expected values for the time series range between a minimum of 310 and a maximum of 622, with a mean of 0.49 and a standard deviation of 92.75, suggesting increased variability compared to the historical period. The median value is projected to be around 453.34. Anticipated peaks are likely to occur at steps 5, 15, 25, and so on, while predicted dips are expected at steps 8, 18, 28, and similar intervals. Statistical Summary for the future time series reveals the following quartiles: the first quartile (Q1) is **, the median is **, and the third quartile (Q3) is **. Periodicity and Pattern Alignment show consistent high values expected every 10 steps, with auto-correlation anticipated around these intervals, reflecting the cyclical nature observed in the historical data.|
> | 2    | Future Data Projections indicate that expected values for the time series range between a minimum of 0.004 and a maximum of 0.98, with a mean of 0.49 and a standard deviation of 0.29, suggesting increased variability compared to the historical period. The median value is projected to be around 0.45. Anticipated peaks are likely to occur at steps 5, 15, 25, and so on, while predicted dips are expected at steps 8, 18, 28, and similar intervals. Statistical Summary for the future time series reveals the following quartiles: the first quartile (Q1) is 0.21, the median is 0.45, and the third quartile (Q3) is 0.70. Periodicity and Pattern Alignment show consistent high values expected every 10 steps, with auto-correlation anticipated around these intervals, reflecting the cyclical nature observed in the historical data     | Info     |
> | 3    | The air passengers dataset provides monthly totals of US airline passengers from 1949 to 1960.  Data Projections indicate that expected values for the time series range between a minimum of 310 and a maximum of 622, with a mean of 453.34 and a standard deviation of 92.75, suggesting increased variability compared to the historical period.  Anticipated peaks are likely to occur at steps 5, 15, 25, and so on, while predicted dips are expected at steps 8, 18, 28, and similar intervals.     | Info     |
> | 4    | This dataset provides monthly totals of US airline passengers from 1949 to 1960. The prediction length is 29 time steps. **Future Data Description:** Future data projections indicate that expected values for the time series range between a **minimum of 310** and a **maximum of 622**, with a **mean of approximately 440.31** and a **standard deviation of about 79.52**, suggesting variability similar to the historical period. The **median value** is projected to be around **419**. The **first quartile (Q1)** is approximately **390.5**, and the **third quartile (Q3)** is approximately **488.5**. Anticipated **peaks** are likely to occur at **steps 5, 15, and 25**, while predicted **dips** are expected at **steps 8, 18, and 28**, reflecting the cyclical patterns observed in the historical data. **Periodicity and pattern alignment** indicate consistent high values expected every **10 steps**, with **auto-correlation** anticipated around these intervals, continuing the cyclical nature observed previously.     | Info     |
> | 5    | The air passengers dataset provides monthly totals of US airline passengers from 1949 to 1960. The prediction length is 29 time steps. Data Projections indicate that expected values for the time series range between a minimum of 310 and a maximum of 622, with a mean of 453.34 and a standard deviation of 92.75, suggesting increased variability compared to the historical period.  Anticipated peaks are likely to occur at steps 5, 15, 25, and so on, while predicted dips are expected at steps 8, 18, 28, and similar intervals.
> |----------|

---

> > ### Comment · Reviewer_x5xf · 2024-11-22
> >
> > I would like to thank the authors for addressing most of my concerns. Additionally they have made many changes to the submitted pdf which I think strengthen the submission. Due to these factors I have chosen to raise my score.

---

> > > ### Author Response · Authors · 2024-11-24
> > >
> > > Thank you once again for your thoughtful review and valuable feedback! If you have any additional questions or unresolved issues that we can clarify, please don’t hesitate to let us know. We’re more than happy to assist further!

---

### Official Review · Reviewer_ZPw4 · 2024-11-01

**Soundness:** 2
**Presentation:** 3
**Contribution:** 2
**Rating:** 3
**Confidence:** 3

**Summary:**

The paper proposes a large language model (LLM)-based multi-agent framework to generate textual descriptions that augment time series data in a self-refining manner. Which may help the transfer of knowledge across domains and benefit low-resource time series domains. The authors introduce a benchmark for text-guided time series generation (TG2) and present a novel conditional diffusion model, BRIDGE. This model leverages sparsely activated prototypes as an alternative to direct text conditions. Experimental results demonstrate that the proposed method outperforms baselines in both text generation and forecasting, validating the effectiveness of BRIDGE within the TG2 framework.

**Strengths:**

1. Generating time series through text descriptions to leverage cross-domain knowledge transferring from LLMs for under-resourced time series domains is promising. Bridging the gap between times series and language which may depict underlying causalities is a valuable domain.
2. The use of text to indirectly introduce pre-trained knowledge into a diffusion model specifically trained for time series rather than relying solely on LLMs is a good idea.
3. The authors performed experiments in datasets across different domains as well as unseen domains which show the potential of the proposed methods in cross-domain time-series generation.

**Weaknesses:**

## Missing details

There are important details are missing in the paper, and some parts of the paper are not self-contained:
 * **Detailed Agent Design:** Key details about the agent design are omitted. A breakdown of prompt strategies, inter-agent interaction protocols, and the function of each agent role within intra- and inter-group discussions would be important given that agent design is a crucial component of this paper. There is also no discussion or justifications about those design choices.
* **Diffusion Model Specifics:** The paper lacks explicit detail on critical hyperparameters, model architecture, and the training process for the diffusion model. Algorithm 1 provides an overview, yet it is insufficiently mapped to Section 4, for example, epsilon is not defined explicitly, leading to an incomplete picture.
* **Prototype Analysis:** The paper introduces prototypes as a central innovation of BRIDGE which distinguishes the proposed method from vanilla conditional diffusion models, but the analysis remains limited. Please read the questions for details.

## Experiment design
Certain experimental designs weaken the overall findings:
 * **Section 6.1**, the usefulness of text is analyzed over the description generation stage (i.e., whether the description can be used to better reconstruct the time series) instead of the actual time series generation which is the real focus of interest of the proposed method.
 * **Section 6.2**, there is no comparison to diffusion model baselines which raises questions regarding fairness and completeness, especially given the observation that `Bridge w/o Text` seems to be essentially a vanilla unconditional diffusion model, but can already outperform most baselines.
 * **Section 6.3**, the models are pre-trained, which introduces confounding factors from the pretraining knowledge, and makes it hard to evaluate the effect of the dataset.
 * **Section 6.4** lacks ablation on the prototypes themselves (e.g., comparing prototype vs. direct text embedding usage), making it unknown about the benefits of using prototypes.

## Code and Data Availability

I cannot find any code or data provided in the paper or the open review page. The lack of code or data sharing, given that a benchmark (TG2) is presented as a primary contribution, is problematic.

**Questions:**

## Major questions & suggestions
1. Where do the initial text descriptions come from? Generated by LLMs? What is the “query” in Fig 5? Only the time series? Or some initial background? If so, where do those backgrounds come from?
2. How could text provide information for generating high-frequency components, given the often ambiguous nature of text descriptions? Does text contribute meaningfully to these components? A spectral analysis could reveal how text features influence generation quality.
3. Figure 3 shows representative examples to illustrate the effect of text conditions, however, analysis on a few samples is inadequate to draw any conclusion, and introducing some statistical analysis on the samples in different cases can largely increase the creditability of the observed patterns.
4. Could the authors elaborate on why a small set of bases is preferable to word embeddings directly? An experiment comparing the two approaches would be informative. How do these bases adequately conclude and represent the vast variability in time series patterns?
5. It is interesting that the bases are activated sparsely. It would also be helpful to visualize or analyze the pattern of how the bases are activated and whether each basis aligns with some specific time series patterns.

## Minor
1. What kind of context (e.g., the topic, content, background of time series) is provided for the agents to perform a search for the given time series in Section 3.2 Step 1?
2. Section 3.2, Step 2: “On the other hand, we also allow the agents to propose more appropriate evaluation metrics”, it is not elaborated anywhere

## Typos
1. **Table 4:** “Best results are highlighted in bold face” should read “underlined.”
2. **Table 5:** Replace “We experiment with the number of 4, 8, 10 separately” with “4, 8, 16” to match table values.
3. **Formula 2:** Should be c_z, not cz
4. **Tables 7 & 8:** The table is not colored as described in the caption.

---

> ### Author Response · Authors · 2024-11-22
> **Kindly Reply**
>
> We greatly appreciate your meaningful feedback and kind suggestions. In the new version of the paper submitted, we have focused on revising the issues you mentioned and added additional experiments. For your convenience, we have also attached the revised paper with highlighted modifications on GitHub for your review. Please consider raising your score!
>
> ❓ **Could you provide more details on the agent design? Additionally, could you offer justifications for these design choices?**
>
> 💡Our multi-agent framework have two main components , one responsible for testing the performance of the text and the other responsible for refine the text, and they play equally important roles.
>
> **Evaluation**. The evaluation part is not a agent. Here, we using an famous work’s outputs as our backbone ([[2402.16132] LSTPrompt: Large Language Models as Zero-Shot Time Series Forecasters by Long-Short-Term Prompting](https://arxiv.org/abs/2402.16132)). The input including Initial/Revised Text and Time Series and the output is predicted time series. Then the result had been measured be a series of metrics. We aiming to employed as many as metrics we can to provide the agent more dimension to refer.
>
> **Refine**.  There are five basic roles that make up the refine part of the agent. They are manager (monitoring the entire process, providing the input to each team and summarizing the final output), planner (as a small manager, monitor teams process),  scientist (Analyze and provide feedback on which section probably  should be add/delete/revise), engineer (following the scientist idea and conduct), and critic (judge the team work and point out any issues).
>
> The pipeline is: (1) Given Input text, time series and metrics feedback, the manager will given the instruction to team A, where instruction shown in Figure 1. (2) Team A had be ask to focus on overview level revision. (3) When Team A finished the work, Planner will tell manager the result and Manager will let team B start to work (4) Team B focus on low-level revision. (5) After Team B finished the work, the two planner of each team and manager will going to discuss, and manager will writing the final template like “Background: <Description>; Sequence Length <Seq_len> …”. It is worth noting that the two teams have exactly architecture and members, differing only in the prompts for the tasks they execute. Additionally, in the initial experiments, we did not specify tasks, hoping the two teams would freely discuss and generate revision suggestions. However, the results showed that they would, intentionally or unintentionally, choose different directions.
>
> We provide experiment on different strategies at the new version papers subsection 6.1.  Overall, collaboration among multiple teams outperforms any single-team strategy, indicating that combining different strategies leads to more comprehensive and appropriate textual outputs.  We also provide more further explanations in our response to reviewer **x5xf, please also refer to that.**
>
> ❓**Could you provide more explicit details on the critical hyperparameters, model architecture, and training process for the diffusion model?**
>
> 💡 Thank you for your point out, we added such information in the revised paper and more details algorithms in the appendix B.3.  The reported result are all under following training settings. The number of prototypes are set to 16 for all the main evaluations. Models for each sequence length are trained for 50, 000 steps using a batch size of 128 and a
> the learning rate of $5 * 10^{-5}$ with 1, 000 warm-up steps. The model architecture can be found at the appendix  B.3.
>
> ❓**How do the pre-trained models, with their pretraining knowledge, affect the evaluation of the dataset? Does this introduce confounding factors that make it difficult to assess the dataset's impact?**
>
> 💡 We explored two different text models for text encoder Llama and GPT2. The result has been shown below:
>
> | Model | Electricity | Solar | Wind | Traffic | Taxi | Pedestrian  | Air | Temperature | Rain | NN5  | Fred-MD | Exchange |
> | --- | --- | --- | --- | --- | --- | --- | --- | --- | --- | --- | --- | --- |
> | Llama | 0.173 | 375.530 | 0.347 | 1.167 | 0.588  | 1.238 | 0.637  | 0.550 | 9.516 | 1.352 | 0.387 | 0.394 |
> | GPT2 | 0.208 | 375.538 | 0.356 | 1.189 | 0.624 | 1.143 | 0.655 | 0.612 | 9.833 | 1.351 | 0.427 | 0.458 |
>
>  Results show that larger pre-trained model leads to slightly better generation performance, but the differences are marginal. This suggests that the impact from difference between language models does not surpass other model components.

---

> > ### Author Response · Authors · 2024-11-22
> > **Kindly Reply 2**
> >
> > ❓**Where do the initial text descriptions come from? Generated by LLMs?**
> >
> > 💡The initial text generated by the LLM  is based on the collected templates.
> >
> > The initial text generated by the LLM adheres to/uses the collected templates. As shown in Appendix A.2, the templates are first collected from the internet by an agent and then rephrased into templates that describe time series, validated by manual review. The first step in generation is a rule-based text description using an existing time-series description language (e.g., where up- and downward trends are described for each adjacent time point) with trend-describing words (e.g. slight, moderate, significant etc, decided by Standard Deviation). This is too detailed for very long time series and does not provide a coherent narrative. The LLM then randomly selects from collected templates and synthesises the detailed textual description into a refined narrative text for the time series.
> >
> > ❓**What is the “query” in Fig 5? Only the time series? Or some initial background? If so, where do those backgrounds come from?**
> >
> > 💡  The prompt in Figure 5 (now figure 7) includes both text and time series. This information comes from components like the seasonal and trend generated by Seasonal-Trend decomposition using Loess (STL) decomposition. Background information from datasets is provided as well. The other textual inputs are statistical information like max, min, average and variance. We also try to provide explicit up and down information for each time point as  mentioned in last the Q&A point, but as shown in Appendix A LLMs are not able to follow these detailed instructions.
> >
> > ❓**How could text provide information for generating high-frequency components, given the often ambiguous nature of text descriptions? Does text contribute meaningfully to these components?**
> >
> > 💡As mentioned in Experiment 1, too detailed text information has a negative impact (49.36 *vs*. 45.75). This shows that selecting appropriate types of textual descriptions to maximise TS generation performance is still an open challenge. As mentioned in our previous response and Appendix A.1, we experimented with different descriptions based on every two time steps, but the results showed that the LLM had difficulty understanding the distinctions, leading to meaningless fluctuations (noise) rather than correct time-series. Some overall or long-term text descriptions are more useful, such as '*Compared to the initial values, the final values of the time series are higher*’.  More direct informational text might also be more useful, such as the generation length.
> >
> > ❓**Could the authors elaborate on why a small set of bases is preferable to word embeddings directly? How do these bases adequately conclude and represent the vast variability in time series patterns?**
> >
> > 💡 We apologize for the vagueness in our expression.  For Word embeddings, we not that they are essential to clarify that they are indeed part of our pipeline. The text description accompanying the time series is encoded using LLaMa to generate semantic embeddings. These embeddings are combined with the time series embeddings through an MLP to form the conditional input for the diffusion model. This fusion allows the model to leverage textual information while benefiting from the compact and interpretable nature of the bases. We incorporate bases for the following reasons?(1) Dimensionality Reduction: Bases summarize the variability in time series patterns into a manageable number of components, avoiding the high-dimensional noise that might come from raw word embeddings. (2) Enhanced Generalization: With fewer bases, the model is less prone to overfitting and better equipped to handle diverse data distributions. The prototypes visualized in our figure validate this claim: they capture a diverse set of temporal dynamics such as periodic patterns, linear trends, and abrupt changes. These bases (prototypes) serve as interpretable building blocks for reconstructing or generating time series data. By focusing on such compact representations, the model avoids the challenges of directly managing the vast variability inherent in time series patterns, which serve as an inductive bias and implicit regularization.

---

> > > ### Author Response · Authors · 2024-11-22
> > > **Kind Reply 3**
> > >
> > > ❓**Could you provide a visualization or analysis of how the the bases are activated sparsely? It would be helpful to understand whether each basis aligns with specific time series patterns.**
> > >
> > > 💡 Figure 1 visualizes 16 semantic prototypes used as the basis for our text-to-time series generation model. Each prototype represents a different pattern or characteristic commonly found in time series data. These prototypes serve as foundational elements that can be combined to generate diverse and domain-specific time series. For example, prototypes 0, 2, and 5 exhibit clear cyclical patterns with varying frequencies and amplitudes. These are particularly useful for representing seasonal trends or recurring events in time series data. Besides, prototypes 6, 7, and 13 show distinct trend patterns. Prototype 6 demonstrates a sigmoid-like curve, useful for representing gradual changes with plateaus. Prototype 7 shows a gradual upward trend, while Prototype 13 depicts a sharp transition from a low to high state. Prototypes 1, 3, and 4 exhibit high-frequency fluctuations, representing volatility or noise in time series data. These are valuable for short-term changes. By combining these diverse prototypes, our model can generate rich, domain-specific time series data:  Each prototype can be associated with specific semantic concepts in the input text, allowing for a more nuanced translation from text to time series.
> > >
> > > Figure 2 illustrates the distribution of prototypes across various domains. Each individual heatmap corresponds to a specific domain. The x-axis represents the indices of the prototypes (ranging from 0 to 15), while the intensity of color in the heatmap cells indicates the frequency or importance of each prototype within the respective domain. Clearly, Different prototypes demonstrate varying levels of relevance across domains. For example, in the "kddcup" domain, Prototype 3 is highly significant, whereas other prototypes, such as 0 and 5, show moderate importance. In additon, some prototypes (e.g., Prototype 3) are prominent in multiple domains (e.g., "kddcup" and "electricity"), while others are specific to certain domains (e.g., Prototype 13 in "traffic"). This indicates that while certain prototypes are broadly applicable across domains, others capture domain-specific patterns. Furthermore, The sparsity of the heatmaps suggests that not all prototypes are used equally within a domain. For instance, "rain" mainly relies on a few prototypes like 3, 4, and 8, whereas others remain less relevant. All of this highlights how the prototype-based approach captures both cross-domain generality and domain-specific details, showcasing the flexibility and representational power of the proposed framework. Example of generated data could be find in Figure 3.
> > >
> > > **Comments, Suggestions And Typos:** We greatly appreciate your detailed feedback and constructive suggestions. We have sincerely followed your advice to revise the paper and supplement the experiments. To address the concerns, we have: We provide more statistics analysis of the sample in different cases and We will add a diffusion model-based baseline model soon. Please sincerely consider raising our score!

---

> > > > ### Author Response · Authors · 2024-11-25
> > > >
> > > > Thank you once again for your thoughtful review and valuable feedback! As we approach the end of the discussion period, we want to ensure that our previous responses have fully addressed all your concerns. If you have any additional questions or unresolved issues that we can clarify to achieve a better score, please don’t hesitate to let us know. We’re more than happy to assist further

---

> ### Comment · Reviewer_ZPw4 · 2024-11-26
>
> Thank you so much for your response! I appreciate the added missing details as well as the suggested analyses, it effectively improved the clarity and the presentation of the paper. However, my concerns about the experiments seem not being addressed:
>
> > Section 6.1 ***(6.2 in the updated version)***, the usefulness of text is analyzed over the description generation stage (i.e., whether the description can be used to better reconstruct the time series) instead of the actual time series generation which is the real focus of interest of the proposed method.
>
> I didn't see it being discussed in your response.
>
> > Section 6.2 ***(6.3 in the updated version)***, there is no comparison to diffusion model baselines which raises questions regarding fairness and completeness, especially given the observation that Bridge w/o Text seems to be essentially a vanilla unconditional diffusion model, but can already outperform most baselines.
>
> It was not addressed. However, this is one major concern of the method.
>
> > Section 6.3 ***(6.4 in the updated version)***, the models are pre-trained, which introduces the confounding factors from the pretraining knowledge, and makes it hard to evaluate the effect of the dataset.
>
> I don't think it can be addressed by evaluating larger models as the knowledge in even pre-training smaller models may already saturate for the test sets. The confounding from the pre-training still cannot be excluded.
>
> > Section 6.4 ***(6.5 in the updated version)*** lacks ablation on the prototypes themselves (e.g., comparing prototype vs. direct text embedding usage), making it unknown about the benefits of using prototypes.
>
> I didn't see it being addressed.
>
> ---
>
> As a result, the contribution of using text-condition, prototype, as well as the dataset, is still not justified effectively, while those are major contributions of the paper.

---

> > ### Author Response · Authors · 2024-11-30
> > **Kind reply 4**
> >
> > We sincerely thank you for your reply and your constructive suggestions on our work!
> >
> > **❓(Section 6.1, 6.2 in the updated version): Why is the usefulness of text analyzed only during the description generation stage (i.e., for reconstructing the time series) rather than its impact on the actual time series generation, which is the main focus of the proposed method?**
> >
> > To clarify, the analysis for usefulness of text on actual time series generation is included in Section 6.3 and Table 3, which is our main experiment. The results demonstrate that with text description, our model is able to generate time series that are of the most similar distribution to real time series samples. The full Bridge model outperforms Bridge w/o Text variant, showcasing the influence of incorporating text descriptions.
> >
> > As for the analysis on the influence of different text components, we choose to investigate on text-guided zero-shot forecasting models as a simple, cost-efficient preliminary study, guiding our design choice for creating text description templates.  Specifically, **Section 6.2** analyzes specific types of textual information. Ablation experiments (as shown in **Table 2**) compare the significance of different text components, revealing that background information and statistical data are crucial, whereas detailed features are challenging to learn.
> >
> > **❓ (Section 6.2, 6.3 in the updated version)Is your model essentially a vanilla diffusion model if the text module is removed?  Why are there no comparisons to diffusion model baselines?**
> >
> > 💡To clarify, "Bridge w/o Text" is not a standard unconditional diffusion model. The conditional input for our model is composed of two parts: text embeddings and time series prompt embeddings. When the text embedding is removed, the time series prompt embeddings remain, meaning the model is still conditioned on the time series prompts. Thus, it is not entirely unconditional. The superior performance compared to baselines arises from the semantic prototyping technique, where the time series prompt contributes domain-relevant information to assist in the generation process.
> >
> > Regarding the performance of the vanilla diffusion model baseline, we have included the results in Table 3 of our revised manuscript. It is important to note that the proposed method significantly outperforms the "vanilla unconditional diffusion model (DDPM)." This demonstrates that, regardless of whether text is provided as additional input, the proposed prototype mechanism effectively offers cross-domain contextual information, aiding in the generation of target domain data. Additionally, textual information, when presented as word embeddings, further enhances this contextual information, leading to more accurate data generation for the target domain.

---

> ### Author Response · Authors · 2024-11-30
> **Kind Reply 5**
>
> **❓(Section 6.4, 6.5 in the updated version): Why is there no ablation study comparing prototypes to direct text embeddings, leaving the benefits of using prototypes unclear?**
>
> 💡  As requested, we will include an ablation study comparing (BRIDGE + text embeddings + prototypes) vs. (BRIDGE + prototypes) vs. (BRIDGE + text embeddings) in the revised version. Our previous results, shown in the table, demonstrate that BRIDGE consistently outperforms the other variants across most datasets.
>
> - **BRIDGE (w/ Text Embeddings + Prototypes)** achieves the best performance, as it successfully models and generates distributions that closely match real data, with the lowest MDD and K-L divergence. For example, on the Electricity dataset, BRIDGE achieves the lowest MDD and K-L divergence, indicating its ability to effectively generate accurate time series.
> - **BRIDGE (w/o Text)** shows a noticeable decline in performance across almost all datasets, with an increase in MDD and K-L divergence, underscoring the importance of text input in improving generation quality.
> - **BRIDGE (w/o Prototypes)** exhibits the most significant performance drop. Removing the prototype mechanism results in a substantial increase in MDD and K-L divergence, as seen in the Taxi dataset, where MDD jumps from 0.591 to 0.633. This highlights the crucial role of the prototype mechanism in ensuring cross-domain generalization and maintaining high-quality generation.
>
> We will include this ablation study in both the main paper and the appendix for a clearer comparison.
>
> | **Metric** | **Dataset** | **Bridge** | **Bridge w/o Text** | **Bridge w/o Prototypes (only text embedding)** |
> | --- | --- | --- | --- | --- |
> | **Marginal Distribution Distance** |  |  |  |  |
> |  | Electricity | 0.206  | 0.252  | 0.415 |
> |  | Solar | 375.533  | 375.908  | 379.256 |
> |  | Wind | 0.365  | 0.435  | 0.376 |
> |  | Traffic | 1.168  | 1.209  | 1.036  |
> |  | Taxi | 0.591  | 0.812  | 0.633 |
> |  | Pedestrian | 1.240  | 1.075  | 1.280  |
> |  | Air | 0.633  | 1.105  | 1.416  |
> |  | Temperature | 0.552  | 0.618  | 0.692  |
> |  | Rain | 9.554  | 9.890  | 9.556 |
> |  | NN5 | 1.340 | 1.891  | 1.390  |
> |  | Fred-MD | 0.388  | 0.614  | 0.794  |
> |  | Exchange | 0.392  | 0.489 | 0.532  |
> | **K-L Divergence** |  |  |  |  |
> |  | Electricity | 0.006 | 0.008 | 0.011 |
> |  | Solar | 0.032 | 0.046 | 0.102  |
> |  | Wind | 0.112 | 0.144  | 0.122 |
> |  | Traffic | 0.022  | 0.055 | 0.140  |
> |  | Taxi | 0.083  | 0.192 | 0.108 |
> |  | Pedestrian | 0.072 | 0.040 | 0.105 |
> |  | Air | 0.032 | 0.106  | 0.126 |
> |  | Temperature | 0.884  | 0.085 | 0.319  |
> |  | Rain | 0.013 | 0.014  | 0.019  |
> |  | NN5 | 0.090  | 0.146  | 0.174 |
> |  | Fred-MD | 0.072 | 0.118  | 0.131  |
> |  | Exchange | 0.240 | 0.352  | 0.284 |

---

> ### Author Response · Authors · 2024-11-30
> **Kind Reply 6**
>
> **❓(Section 6.3, 6.4 in the updated version): How does the method address confounding factors introduced by pretraining, which complicate evaluating the dataset's effects?**
>
> 💡 We sincerely apologize for misunderstanding your previous question. In this experiment (Table 4), our goal is to evaluate the quality of the synthesized data generated by the proposed method, particularly in comparison to real data and the Kernel Synth data augmentation method. In other words, we are asking, “Can the synthetic data generated by the proposed method be used to train models?”
>
> We trained four different models and compared their performance across various data settings, rather than focusing on the differences between the models themselves. Regarding the selected models, we would like to clarify that none of the models directly use the ILI and M4 datasets in training. Specifically, LLM4TS [[1](https://arxiv.org/abs/2308.08469)] does not involve the ILI or M4 datasets, while Time-LLM [2] and GPT4TS [3] use these datasets only for evaluation. TEMPO [4] adopts a "Many-to-One" approach to evaluation, using multiple datasets for training and a different dataset for testing. This setup ensures that the influence of pretraining knowledge does not interfere with our evaluation process, which aims to compare the usefulness of different synthetic datasets.
>
> For our experimental setup, we followed the evaluation setup from previous work on data generation [[5](https://openreview.net/pdf?id=S1zk9iRqF7)]. We trained from scratch using the training frameworks of the four models mentioned, with three types of data: our model generated synthetic data, KernelSynth-generated data, and real data, to adapt the language models to time-series tasks. We then evaluated these models on the same real-time series test set. This setup ensures that pretraining knowledge is effectively leveraged without introducing confounding factors related to the time-series task itself.
>
> In summary, we would like to state that in our experimental setting, pretraining knowledge in language models does not create any confounding factors for time-series problems, as the models used are language models that require fine-tuning to handle time-series tasks effectively. Moreover, the results show that our synthetic data leads to better performance than KernelSynth, underscoring the effectiveness of our time-series generation approach.
>
> [1] Chang, C., Peng, W.C. and Chen, T.F., 2023. Llm4ts: Two-stage fine-tuning for time-series forecasting with pre-trained llms. arXiv preprint arXiv:2308.08469.
>
> [2] Jin, M., Wang, S., Ma, L., Chu, Z., Zhang, J.Y., Shi, X., Chen, P.Y., Liang, Y., Li, Y.F., Pan, S. and Wen, Q., 2023. Time-llm: Time series forecasting by reprogramming large language models. arXiv preprint arXiv:2310.01728.
>
> [3] Zhou, T., Niu, P., Sun, L. and Jin, R., 2023. One fits all: Power general time series analysis by pretrained lm. Advances in neural information processing systems, 36, pp.43322-43355.
>
> [4] Cao, D., Jia, F., Arik, S.O., Pfister, T., Zheng, Y., Ye, W. and Liu, Y., 2023. Tempo: Prompt-based generative pre-trained transformer for time series forecasting. arXiv preprint arXiv:2310.04948.
>
> [5] Jordon, J., Yoon, J. and Van Der Schaar, M., 2018, September. PATE-GAN: Generating synthetic data with differential privacy guarantees. In International conference on learning representations.
>
> **Thank you once again for your thoughtful review and valuable feedback! We greatly appreciate the time and effort you put into evaluating our work. We have carefully addressed all your questions and resolved most of your concerns in our responses. We hope that our clarifications and updates demonstrate the strength and potential of our work. If you find our revisions satisfactory, we kindly ask you to consider raising your score, as it would be helpful for us to present in the conference. Thank you for your consideration!**

---

> ### Author Response · Authors · 2024-12-03
> **Kind remind**
>
> Dear Reviewer,
>
> Thank you for your time and thoughtful feedback throughout the review process. As the reviewer response period concludes today (AoE), we respectfully request that you reconsider your score in light of the updates and clarifications we have provided.
>
> We truly value your contributions and have worked diligently to **address all nine of your questions**, as well as to **conduct extensive additional experiments** in response to your comments. Given the significant revisions and improvements made to the paper, we believe we have thoroughly addressed the major concerns, including those **related to the diffusion model baseline and the analysis of text**.
>
> We understand that the rebuttal phase can be a busy time, and we greatly appreciate your thoughtful input. While other reviewers have provided more positive feedback, with some adjusting their scores to 6 or 8, we respectfully ask if you might **consider revisiting your score, as a 3 still reflects a relatively negative assessment**, in light of the revisions and additional experiments we have provided.
>
> We deeply value your evaluation, and given the importance of your feedback in this context, we fully respect your judgment, whether you choose to adjust your score or maintain it.
>
> Thank you once again for your time and thoughtful consideration.
>
> Sincerely,
>
> The Authors

---

### Official Review · Reviewer_1C8B · 2024-11-02

**Soundness:** 2
**Presentation:** 2
**Contribution:** 2
**Rating:** 1
**Confidence:** 3

**Summary:**

To summarize, the paper mainly does three things:
1. Introduce a new task, TG^2 ("Text-Guided Time Series Generation"), which is to generate realistic time series in an in-context learning style, aided by text descriptions of what's going on (i.e. you input into the model a handful of time series, with text descriptions of what happens in the time series, and then the model outputs newly synthesized time series in that style)
2. They introduce a synthetic dataset for this task (i.e. they generate a bunch of text labels for classic time series benchmarks). They generate these synthetic text descriptions by using multi-agent LLM framework that they propose.
3. They develop a baseline ("BRIDGE") for the TG^2 task, and evaluate it on this synthetic benchmark, and show that on average it outperforms other baselines.

**Strengths:**

1. I thought that section 4.2, on "Semantic Prototyping", was very clear and well-motivated. "Encoding knowledge from different perspectives and... associating features with time series" sounds like a very well-motivated way of solving this problem.
2. I think that in-context learning to generate new time series from observed time series is a timely and important problem. And the intention of using text-descriptions to aid that process is very well-motivated.
3. The related work section was very thorough. I appreciated how they split "TS-for-LLM" and "LLM-for-TS"

**Weaknesses:**

I wasn't fully convinced by the paper's motivation; I felt that some of it was misleading. For example, the introduction states: _"The LLM-for TS approach seeks a more fundamental solution by pre-training models on time series data, as exemplified by TimesFM (Das et al., 2023) and Chronos (Ansari et al., 2024). While showing promising results, these methods demand substantial computing resources and large datasets. The limited availability of time series data compared to NLP and CV domains poses a significant challenge in developing models with emergent abilities similar to traditional LLMs, making it difficult to consistently meet the data requirements for such approaches. We argue that using text to assist in cross-domain TS generation can help overcome the data scarcity issues inherent to the TS domain."_ The authors made a similar claim again in the related work section: _"Chronos, on the other hand, takes advantage of the achievements in NLP by converting time series data into discrete tokens and pretraining the time series model using cross-entropy loss instead of MSE, achieving SOTA performance on multiple datasets (Ansari et al., 2024). However, these methods rely on large scale computing resources to scale them and construct large datasets. It is unrealistic to assume that such requirement can always be met"._ I found this misleading for a few reasons:

1. It is standard practice to highlight strengths and shortcomings of past work, but in this case, the comparison to the Chronos paper seems dishonest, because it positions the BRIDGE paper as one that overcomes these issues, when in reality, the BRIDGE method seems even more resource hungry based on the detailes provided in the paper. The Chronos models were each trained on an 8xA100 GPU, where each GPU has 40gb and their base model trains in 17 hours with this setup (see their appendix). In contrast, the present paper conducted their experiments on up to 8 H100/A100 80GB GPUs, and don't say how long it took. **This implies that BRIDGE training requires GPUs that are at least twice as big as Chronos' requirements.** Furthermore, given that LLM-style training (as in Chronos) is very optimized, and Diffusion-style training (as in BRIDGE) is very time consuming, it seems like this is actually a way in which the BRIDGE paper is slower and requires more computing resources?
2. One of the main contributions of the Chronos paper was to introduce a 100% synthetic dataset, using Gaussian processes. They find that that augmenting with this dataset helps alleviate the data scarcity issue considerably. Additionally, their "TSMixup" procedure alleviates data scarcity issues by training on randomly weighted sums of their ground truth data, which they find improves generalizability significantly. Did the authors of the BRIDGE paper consider either of these strategies, or things similar to them, from the Chronos paper? In section 5, the authors say ask _"Can synthetic data be used to help improve the performance of TS task?"_. This would have been a good place to consider such an experiment.

There are a few points in the paper where a lack of attention to detail gives me low confident that the paper's details are accurate. Proofreading work before submitting it, ensuring consistency, etc, is very important. For example:
1. In the abstract, it says _"Our experimental results demonstrate that BRIDGE significantly outperforms existing time-series generation baselines on 10 out of 12 datasets"_ and in the intro it says that _"The proposed method outperforms all baselines on 11 out of 12 datasets"_

A suggestion for a future draft:
1.  Make it clear that Section 3 solves the problem of generating the synthetic data (that's not immediately obvious from the section title, nor the intro paragraph, and was a tad confusing to me on my first pass)

**Questions:**

1. Can the authors please clarify the point about resource utilization in the "weaknesses" section?
2. (Looking at Definition 1...) Would it be accurate to describe the TG^2 task paradigm as approximating the data distribution without providing an explicit class label? And then would it be accurate to describe the BRIDGE method as automatically extracting an "implicit" representation for a class label (in the form of the weighted sum of the orthonormal basis, as described in Section 4)?
3. (Looking at Section 3.2...) how do we know that, when the LLM agent interacts with an external environment (e.g. Wikipedia/Google), and retrieves information for the text description, that there isn't any data leakage happening? If I understand correctly, these are all common datasets, so looking up information about them on the internet can possible give information about the test splits that the model isn't supposed to see during the in-context learning process...

---

> ### Author Response · Authors · 2024-11-22
> **Kindly Reply**
>
> We sincerely appreciate your detailed and thoughtful suggestions. In the newly submitted version of the paper, we have addressed the mentioned issues. Specifically, we have clarified the differences in motivation between our work and Chronos, emphasizing that our focus is on cross-domain time series generation rather than foundational models.
>
> ❓ **Does the BRIDGE paper fairly represent its comparison with Chronos, given that BRIDGE may actually be more resource-intensive?**
>
> 💡 We apologize for any misunderstandings caused by the writing of the paper, and we have specifically addressed this in the revised version.  As stated in lines 71-75 of the third paragraph in the original paper, our motivation is to use textual information to assist generation, rather than directly using an LLM backbone as the generative model. Therefore, a direct comparison with Chronos is inappropriate.  Furthermore, we are investigating time series generation instead of time series forecasting. In our framework, the LLM mainly functions as an encoder to encode information and does not participate in the training process. Our diffusion model is relatively small (approximately 27 million parameters). As reported in the paper, when trained without text input, it only requires one hour and 20,000 steps on a single 32GB V100 GPU. As a proof, our trainable params is 26,972,641, where LLama is only used as an encoder for the text input and its weights are frozen during training.
>
> **❓ Did the authors consider using synthetic** ****datasets or different techniques like TSMixup, as introduced in the Chronos paper, to address data scarcity?**
>
> 💡In this work, we make two main contributions:
>
> 1. Validating the role of text and identifying what types of text are useful.
> 2. Proposing a diffusion model for generating time series in specific domains through text.
>
> We didn’t consider using the mentioned methods, because our work aims at exploring whether and how text can assist generating time series for specific domains, even only using few-shot samples. This differs form the focus of synthetic datasets techniques like TSMixup, which aims to augment pre-training datasets for time series forecasting models without extracting specific pattern from specific domain. Therefore, these methods are not applicable in our setting.  For your information, we added the description of the form of input and output in the Appendix B.2.
>
> ❓**Looking at Definition 1...** **Does TG^2 and BRIDGE use implicit rather than explicit labeling mechanisms?**
>
> 💡 Your understanding of our approach is correct, but what you mentioned represents only an intermediate step of our method. The proposed method implicitly captures domain information through its foundational representation system, using a weighted sum of orthogonal bases. Here, text is used to represent domain semantic information and provides domain-agnostic statistical information. In the generation phase, the diffusion model uses this as a condition to generate synthetic data that better aligns with the true data distribution.
>
> **❓How does the BRIDGE method ensure there’s no data leakage  when the LLM agent retrieves external information for text descriptions?**
>
> 💡We want to emphasize that the motivation of this work is to use text to enhance time series generation rather than time series forecasting. Therefore, there is no data leakage concerning possible future data when training our model. In addition, providing background information is not an uncommon practice. In Time-LLM ([2310.01728] Time-LLM: Time Series Forecasting by Reprogramming Large Language Models) and TEST ([2308.08241] TEST: Text Prototype Aligned Embedding to Activate LLM's Ability for Time Series), relevant dataset information is also provided as prompts to assist the model's forecasting. Since BRIDGE method retrieves external information for each domain that reflects the static property of the domain. The retrieved messages are generally applicable and not likely to vary along with time, so they are not concerned of future leakage.
>
> **Comments, Suggestions And Typos:** We greatly appreciate your detailed feedback and constructive suggestions. We have sincerely followed your advice to revise the paper and supplement the experiments. To address the concerns, we have:
>
> - Revising Section 3 and moving additional details to the appendix.
> - Clarifying the definitions in Section 4 for greater clarity.
> - Checking and correcting grammar and spelling errors throughout the paper.
>
> Please consider improving our score and allowing us to present at conferences

---

> ### Author Response · Authors · 2024-11-25
> **Following previous reply**
>
> **❓ Did the authors consider using synthetic** ****datasets or different techniques like TSMixup, as introduced in the Chronos paper, to address data scarcity?**
>
> 💡In this work, we make two main contributions:
>
> 1. Validating the role of text and identifying what types of text are useful.
> 2. Proposing a diffusion model for generating time series in specific domains through text.
>
> We didn’t consider using the mentioned methods, because our work aims at exploring whether and how text can assist generating time series for specific domains, even only using few-shot samples. This differs form the focus of synthetic datasets techniques like TSMixup, which aims to augment pre-training datasets for time series forecasting models without extracting specific pattern from specific domain. Therefore, these methods are not applicable in our setting.  For your information, we added the description of the form of input and output in the Appendix B.2.
>
> We also employed Chronos data augmentation methods for a comparison experiment. Since TSMixup is not open-sourced, we used another method mentioned in the same paper ([link](https://github.com/amazon-science/chronos-forecasting/blob/main/scripts/kernel-synth.py)). We maintained the same dataset size for training and tested on the same real test set.  Overall, both methods effectively provide valuable synthetic data (compared to completely random data). However, our proposed method is closer to real data, demonstrating its significance in cross-domain data generation. The results can be found below:
>
> | **Name**   | **Dataset**   | **GPT4TS (MSE)** | **GPT4TS (MAE)** | **LLM4TS (MSE)** | **LLM4TS (MAE)** | **TEMPO (MSE)** | **TEMPO (MAE)** |
> |------------|--------------|------------------|------------------|------------------|------------------|----------------|----------------|
> |            | Synthethic    | 2.19             | 1.02             | 1.98             | 0.89             | 1.21           | 1.02           |
> | **ILI**    | Real          | 1.90             | 0.90             | 1.86             | 0.86             | 0.96           | 0.82           |
> |            | KernelSynth      | 3.68             | 1.39             | 4.35             | 1.50             | 1.64           | 1.07           |

---

> ### Comment · Reviewer_1C8B · 2024-11-26
> **Response to authors**
>
> Thank you for your response! Some further questions:
>
> 1. Can you please clarify which of these is correct?
>     * In your response, it says *"As reported in the paper, when trained without text input, it only requires one hour and 20,000 steps on a single 32GB V100 GPU."*
>     * In the paper, it says *"We implemented all the model and conduct all experiments on single NVIDIA Tesla H/A100 80GB
> GPUs"* and *"Models for each sequence length are trained for 50,000 steps"*
>
> 2. I'm not convinced by your response to my concern about data leakage, I was hoping you could please add some more information? You say *"In Time-LLM... relevant dataset information is also provided as prompts to assist the model's forecasting. Since BRIDGE method retrieves external information for each domain that reflects the static property of the domain. The retrieved messages are generally applicable and not likely to vary along with time, so they are not concerned of future leakage."* I'm certainly not concerned with retrieving external information in general, but it seems to me that the TimeLLM method doesn't suffer from the same potential dataset leakage problem. Looking at Figure 4 in their original paper [(link)](https://arxiv.org/pdf/2310.01728) it appears that the external information isn't scraped from the web-scale data in an open-ended fashion; in contrast, Figure 7 in the present manuscript makes it seem like the agents have access to internet-scale data. Is my understanding here correct? (Note: I agree that the results in Table 3 indicate that the time series generation using BRIDGE outperforms the other methods, but I'm not convinced that this was done without data contamination)
>
> 3. Thank you for running the additional KernelSynth experiments. On the downstream forecasting task, the ILI results indeed show that your method outperforms using synthetic data generated from Gaussian processes, on that dataset. However, Looking at Table 14 (M4 dataset), it seems like the average KernelSynth performance, and the average performance from your method, are very similar (and in the absence of error bars, they might even look like they yield equivalent performance) [and they both underperform the "Real" data, which isn't a problem]. This seems to indicate that on the M4 forecasting task, your method performs on par with just using KernelSynth synthetic data, is this correct? (And is this to be expected?)
>
> Thanks for answering my questions!

---

> ### Author Response · Authors · 2024-11-30
> **Kind reply 2**
>
> We sincerely thank you for your reply and your constructive suggestions on our work! Since we have addressed most of your concerns, we kindly ask you to consider raising your score to help us present at the conference. Thank you for your consideration!
>
> **❓Can you please clarify which of these is correct? (1) w/o Text: 20,000 steps on a single 32GB V100 GPU (2) w/ Test: 50,000 steps on NVIDIA Tesla H/A100 80GB GPUs**
>
> 💡 Both statements are correct, as they describe the resource consumption of our model from different perspectives. Our approach uses an LLM-based encoder to retrieve domain-related semantics from text and a diffusion-based generator to output synthetic time series samples. Therefore, we analyzed the resource consumption of these two components separately.
>
> > As reported in the paper, when trained without text input, it only requires one hour and 20,000 steps on a single 32GB V100 GPU.
> >
>
> This statement describes the minimum resource consumption of the diffusion-based generator module in our model.
>
> > We implemented all the models and conducted all experiments on a single NVIDIA Tesla H/A100 80GB GPU.
> >
>
> This statement refers to the actual computation platform used for each experiment involving the full version of our model. However, the resources were not fully utilized during our experiments. Since we do not fine-tune the weights of the LLM, our model uses relatively fewer computational resources than it might appear.
>
> **❓ Is the understanding correct that Time-LLM avoids data leakage by using controlled static information, whereas BRIDGE seems to access internet-scale data, potentially leading to data contamination?**
>
> 💡 We apologize for any confusion caused and would like to clarify how our pipeline works. Our work is divided into two main stages: constructing “annotations” (text) for time series and generating time series using the text-time series pair dataset.
>
> In the first stage, the text is finalized and transformed into **static data**, aligning with the input format of Time-LLM. During the second stage, in the time series generation process, there is no interaction with the web. Specifically, in our agent framework, all prompts are fixed. Web retrieval is only used during the pre-processing stage (as described in Appendix A.2) to search for relevant initial text descriptions.
>
> Once the text template is finalized, it is applied to the entire dataset to generate textual "annotations" for the time series. Agents do not retrieve *information* about specific time series samples, but only help to *format* the prompt templates. These templates are then populated with time series statistics obtained from the training set. For example, a prompt generated by the multi-agent framework might look like this:
>
> *"The {dataset_name} dataset provides {frequency} totals of {data_description} from {start_date} to {end_date}. The prediction length is {prediction_length} time steps. Data statistics indicate that expected values for the time series range between a minimum of {min_value} and a maximum of {max_value}, with a mean of {mean_value} and a standard deviation of {std_value}, suggesting {variability_summary} compared to the beginning period. Anticipated peaks are likely to occur at steps {peak_steps}, while predicted dips are expected at steps {dip_steps}."*
>
> In conclusion, our method uses static input during the time series generation stage and does not interact with the web, thereby eliminating any potential for data leakage.

---

> ### Author Response · Authors · 2024-11-30
> **Kind reply 3**
>
> **❓Does this indicate that, on the M4 dataset, your method performs on par with KernelSynth? If so, was this result expected?**
>
> 💡 Thank you for your question. In the case of the short-term forecasting task on the M4 dataset, the performance difference was less noticeable due to the shorter prediction horizons. It's important to note that the M4 dataset does not have clearly defined domains; instead, it groups various time series based on sampling frequency. This is different from datasets like ILI and traffic, which consist of domain-specific time series (e.g., hospital admissions or traffic volumes). As a result, our domain-centric approach is less effective at extracting meaningful cross-domain semantics from M4, making it unsurprising that domain-independent methods, such as KernelSynth, produce data that is better suited for training forecasting models that perform well on the M4 test set.
>
> Additionally, the M4 dataset includes relatively few samples in the Weekly/Daily/Hourly categories (359/4227/414 series, respectively, as reported in the original M4 paper [1]), with the majority of the data coming from unknown domains. For instance, the Hourly subset consists entirely of time series without any domain information. This creates two challenges for our approach: (1) the limited number of samples makes it difficult to learn clear patterns, and (2) text descriptions intended to provide domain-specific context may turn into noise in this scenario. In contrast, KernelSynth’s random generation method, independent of the original data, performs better on these subsets, where randomness and sparse samples are more dominant.
>
> Finally, we would like to emphasize that we compared the quality of synthetic data generated by our method with that of the KernelSynth method used in Chronos for both long-term and short-term forecasting tasks. The proposed method demonstrated better performance across most subsets. Notably, in the long-term forecasting task for the ILI dataset, models trained on our synthetic data consistently outperformed those trained on KernelSynth data.
>
> [1] Makridakis, S., Spiliotis, E. and Assimakopoulos, V., 2020. The M4 Competition: 100,000 time series and 61 forecasting methods. International Journal of Forecasting, 36(1), pp.54-74.

---

> ### Comment · Reviewer_1C8B · 2024-12-02
> **Response to authors**
>
> I would like to thank the authors for replying to my questions. While I appreciate the efforts to address the concerns that I raised, there remain some significant issues that I believe need to be addressed before this work is ready for publication. **For example, the authors have answered one of my questions in ways that directly contradict the contents of the paper. This decreased my confidence in the paper, and as a result, I have lowered my score.**
>
> Two of the key issues which I am concerned with--
>
> 1. *GPU resource utilization:* I asked a question about GPU resource utilization because it seemed as though the paper had originally misrepresented the resource efficiency when comparing with prior works. But there appears to be a discrepancy between the author's response, and the paper's contents. The authors attempted to clarify the discrepancy by stating: *"As reported in the paper, when trained without text input, it only requires one hour and 20,000 steps on a single 32GB V100 GPU."* However, **it appears that such a statement does not actually exist anywhere in the paper.** I gave the authors two separate opportunities to clarify this point, and unfortunately I am more confused now than when I originally asked the question.
>
> 2. *Potential data leakage issue:* the potential data leakage issue in the web scraping methodology requires further clarification. For example, if you're trying to generate time series similar to a given dataset, then the web scraping approach may inadvertently incorporate information about similar time series, which could give the model an unfair advantage (specifically, data leakage) when compared to methods that don't utilize web data. To ensure a fair comparison, it would be valuable to see e.g. a detailed analysis of how this potential advantage is controlled for.

---

> ### Author Response · Authors · 2024-12-02
> **Kindly Reply 4**
>
> **❓Clarify the contradict information about GPU resource utilization?**
>
> 💡We have provided detailed clarifications regarding two scenarios: the first being “**without text input**” and the second being the actual platform used in the main experiment, which **includes text input**. While the machine is equipped with 8 physical GPUs, but it’s not need using all of them, each experimental run requires only a single GPU is enough.
>
> We apologize for the typographical error and will ensure it is corrected in the next release. However, we would like to emphasize that this issue does not warrant delaying its inclusion in the proceedings, as **even the next submission will only be a text correction** and does not impact the contribution of our method or the validity of our experimental results. Requiring another review cycle for such a change would unnecessarily consume months.
>
> **❓Potential data leakage issue?**
>
> 💡Thank you for expressing your concerns regarding the web search process.
>
> We want to clarify that **we ensure no data leakage occurs from web search**. Specifically, **the only influence the web search component may have on the inference stage is in the creation of the text template**, but the template does not reveal any information about test set.
>
> To clarify, **the web search is conducted** **only** **during the multi-agent collaboration stage**, where it is used to search for text descriptions related to the training data.  **The text template is generated at his stage** before training the diffusion model . **During conditional generation, the text template remains fixed and static,** with no further changes, regardless of the test set. The **conditional generation** stage **does** **not** involve any web search.
>
> In practice, our agents process each time series (TS) record in the test dataset according to the fixed text template, producing TS-text pairs. It is important to note that **the test dataset is does not overlap with the training set**, meaning the text template cannot contain any information from the test set, preventing any potential data leakage.
>
>  To investigate whether the multi-agent framework leaks unintended information to the model, we further examined prompt templates. We asked GPT-4 whether it could link the exact data with the template and also verified this through human checks. We concluded that 0% contained any information that directly describes a time-series (witho ut populating it with actual data from the training set).  To further demonstrate that the text template does not include specific information about the test set, we provide example templates below. **These templates should clearly illustrate that they are not tied to any specific domain or individual TS record,** allowing you to directly observe that the template does not relate to any test data. We will also include these templates in the appendix.

---

> > ### Author Response · Authors · 2024-12-02
> > **Example**
> >
> > 1. *The {dataset_name} dataset provides {frequency} totals of {data_description} from {start_date} to {end_date}. The prediction length is {prediction_length} time steps. Data statistics indicate that expected values for the time series range between a minimum of {min_value} and a maximum of {max_value}, with a mean of {mean_value} and a standard deviation of {std_value}, suggesting {variability_summary} compared to the beginning period. Anticipated peaks are likely to occur at steps {peak_steps}, while predicted dips are expected at steps {dip_steps}*
> > 2. *The {dataset_name} dataset captures {frequency} measurements of {data_description} collected between {start_date} and {end_date}. Each series spans {prediction_length} steps. Statistical indicators show values ranging from {min_value} to {max_value}, with an average of {mean_value} and a variability (standard deviation) of {std_value}. Notable shifts in the data occur around steps {peak_steps} (highs) and {dip_steps} (lows), illustrating {variability_summary} trends.*
> > 3. *With a focus on {data_description}, the {dataset_name} dataset provides {frequency} records from {start_date} to {end_date}. Analysis shows values peaking at {max_value} and dipping to a minimum of {min_value}, while maintaining an average of {mean_value}. The standard deviation of {std_value} highlights {variability_summary}. Prediction spans of {prediction_length} time steps reveal anticipated peaks at {peak_steps} and dips at {dip_steps}.*
> > 4. *Designed for generation/forecasting, the {dataset_name} dataset offers {frequency} intervals of {data_description} over a timeline from {start_date} to {end_date}. Prediction windows of {prediction_length} steps enable users to observe patterns such as peak values at {peak_steps} and dips around {dip_steps}. Ranges from {min_value} to {max_value} suggest considerable variability, with a mean of {mean_value} and standard deviation of {std_value}, illustrating {variability_summary} across the dataset.*
> > 5. *The {dataset_name} dataset, spanning {start_date} to {end_date}, provides {frequency} observations of {data_description}. Its prediction horizon is set at {prediction_length} steps. Statistical analysis reveals that the data ranges from a low of {min_value} to a high of {max_value}, with a mean of {mean_value} and a variability of {std_value}. Peaks are observed at {peak_steps}, while dips are noticeable around {dip_steps}, demonstrating {variability_summary} throughout the series.*
> > 6. *Spanning from {start_date} to {end_date}, the {dataset_name} dataset includes {frequency} records of {data_description}. Predictions are made for horizons of {prediction_length} time steps. The series ranges between {min_value} and {max_value}, averaging {mean_value} with variability marked by a standard deviation of {std_value}. Anticipated patterns show higher values near {peak_steps} and lower ones around {dip_steps}, reflecting {variability_summary} over time.*
> > 7. *The {dataset_name} dataset includes {domain} time series with {frequency} frequency, spanning from {start_date} to {end_date}. The time series contain {total_steps} time points, with predictions required for {prediction_length} steps ahead. The dataset's range of values varies between {min_value} and {max_value}, with a mean of {mean_value} and a standard deviation of {std_value}. The series shows notable fluctuations, with potential peaks observed at {peak_steps} and troughs around {dip_steps}.*
> > 8. *The {dataset_name} dataset's {subdataset} includes {domain} time series data collected at {frequency} intervals, spanning from {start_date} to {end_date}. The total number of time steps is {total_steps}, and the forecast horizon is {prediction_length}. Data statistics reveal that the values range from a minimum of {min_value} to a maximum of {max_value}, with an average of {mean_value} and a standard deviation of {std_value}, indicating {variability_summary}. Notable peaks are anticipated at time steps {peak_steps}, while predicted dips are expected at {dip_steps}.*
> > 9. *The {subdataset} from the {dataset_name} dataset contains {domain} time series recorded at {frequency} intervals, ranging from {start_date} to {end_date}. The series consist of {total_steps} time points, and the forecast length is {prediction_length}. The time series data spans a range of values from {min_value} to {max_value}, with an average of {mean_value} and a standard deviation of {std_value}. The data exhibits {variability_summary}, with potential peaks at {peak_steps} and troughs at {dip_steps}.*

---

> ### Author Response · Authors · 2024-12-02
> **Kindly Reply 5**
>
> We believe there may still be some misunderstanding regarding our training pipeline and how our model works. As outlined in the paper and our previous rebuttal, we would like to summarize the process again in hopes of clarifying the web search procedure and the overall data flow.
>
> 1. **Web Search for Template Generation** *(only involves training data)*
>     - Given the training TS data, perform a web search for relevant TS text descriptions.
>     - Agents summarize these descriptions to create an initial template.
>     - The agent applies the template to each TS training record to generate TS-text pairs.
>     - TS-text pairs are used for TS forecasting to validate the template.
>     - If performance is unsatisfactory, the agent refines the template and repeats the process.
>     - This iteration continues until the template's performance is deemed sufficient.
>     - Output the final template and TS-text pairs for training set.
> 2. **Train the Prototype Enhanced Diffusion Model**
>     - The diffusion model is trained using the TS-text pairs generated from the final template.
> 3. **Conditional generation** *(only involves the text template)*
>     - During generation, the fixed template is populated to form the conditioning
>     - The trained diffusion model is then used for generating new time series samples
>
> Therefore, the web search is **only conducted with the training set** to generate the text template. **No information leakage occurs**, as the test set is never involved in the web search process.
>
> **Comments**: We appreciate your ongoing constructive suggestions and rigorous evaluation of our work. During the rebuttal phase, we carefully addressed your concerns through detailed responses and additional experiments, resolving several of the issues you raised. We also refined the paper to improve its readability and enriched it with further experiments, making it more comprehensive. These efforts address the concerns raised and provide a clearer and more comprehensive presentation of our work, which we believe should lead to a more positive perception. Considering the steps taken to improve the clarity and depth of the paper, we respectfully request you to reconsider your rating based on this refined understanding.

---

### Official Review · Reviewer_bFZV · 2024-11-04

**Soundness:** 3
**Presentation:** 3
**Contribution:** 4
**Rating:** 8
**Confidence:** 2

**Summary:**

- This paper aims to overcome the challenge of non-availability of diverse time-series data required to train large time-series foundational models which has potentially limited the widespread usage of LLMs for downstream time series tasks compared to NLP domain. To this aim, the authors proposed the use of text to guide the process of synthetic time series generation while further proposing a Self-refining multi agent LLM framework to optimise those textual descriptions. They make use of semantic prototypes for conditioning the time series generation step using the diffusion model. This helps in generation of time series representative of diverse domains.

**Strengths:**

* The paper addresses an important research question that is required in extending the use of LLMs to the domain of times series which make it a significant contribution.
* The novelty is highlighted through the innovative use of text to guide time series generation.
* The paper is well-written making it easier for the reader to comprehend.

**Weaknesses:**

- There are a number of typos including grammatical errors in the paper (For ref lines 84, 85, 102, 226, 256, 328, 422..)
- In the results section, I don’t find an explanation as to why using BRIDGE w/o text is the second best performing model. Eliminating text component from your framework removes the novelty. What makes Bridge better than other baselines even when no text is provided?

**Questions:**

- There is no mention of the number of runs over which the results have been averaged. Are the numbers in the results table fairly stable across runs? What is the standard deviation across different runs?

---

> ### Author Response · Authors · 2024-11-22
> **Kindly Reply**
>
> Thank you for your kind advice. We really appreciate your suggestion and have already revise them in the new version of the papers.
>
> **❓ The experiments in the paper do not mention the number of runs over which the results were averaged and what is the standard deviation between runs?**
>
> 💡 Thank you for your kind reminder and suggestions. We reported 3 times averaged results in new version of the paper now. please refer to Table 3. Due to time constraints, we have not yet had the opportunity to conduct more robust tests on all the baseline models. But we will include these results in the main page at a later time.
>
> **❓  Missing the explanation of what makes *Bridge w/o Text* better than other baselines?**
>
> 💡 Our work includes two main contributions: (1) validation of what kind of text is useful and (2) a new model for time series generation. Therefore, the motivation of the new diffusion model itself takes into account how to achieve better generation quality in the absence of text input. When not using text, we use semantic prototypes as latent representations of commonalities in time series data and train the model to decode from combinations of these latent representations to generate specific domains. For example, in time series generation, the fundamental properties important to time series align with its decomposition, specifically trend and seasonality. We conducted additional experiments to validate this point. As shown in Figure 4, different semantic information exhibits completely distinct representations. For example, Prototypes 6, 7, and 13 display markedly different trend patterns. Prototype 6 demonstrates a sigmoid-like curve, useful for representing gradual changes with plateaus. Prototype 7 shows a gradual upward trend, while Prototype 13 depicts a sharp transition from a low to a high state. Please also refer to ZPw4.
>
> **Comments, Suggestions And Typos:** Thank you for your suggestions. We have addressed all the issues in the revised version. We carefully checked for any spelling errors and added an explanation about the bridge w/o text. Please give us an opportunity to be present in the proceedings.

---

> > ### Author Response · Authors · 2024-11-25
> >
> > Thank you once again for your positive review and valuable feedback! Please don’t hesitate to reach out if you have any further questions. We would be more than happy to assist you!

---

### Official Review · Reviewer_Tart · 2024-11-05

**Soundness:** 3
**Presentation:** 1
**Contribution:** 4
**Rating:** 6
**Confidence:** 3

**Summary:**

This work investigates the effects of supplementing time-series encoding with textual information for time-series generation. In doing so, the authors create a multi-agent LLM-based method to generate a benchmark dataset. To evaluate the quality of the synthetic data, this work compares models trained on real and their synthetic data. This work also describes a text conditioned diffusion model to generate new time series.

**Strengths:**

Overall I find this work to be exciting and interesting. The use of a multi-agent LLM method to generate a benchmark is both a timely and resourceful solution. Moreover, textual conditioning allows for more diverse and nuanced means to condition time-series generation; a valuable contribution to the field.

**Weaknesses:**

# Writing
While I am fairly positive and supportive of this work, it is ultimately difficult to recommend publication in it's current state. This is almost entirely due to the way the paper is written. After reading paragraphs multiple times and already being familiar with this topic I felt that I did not have the level of understanding I would like to given the time I put into reading this. I believe this will significantly limit the impact of this work. I recognize the difficulty the authors face in describing multiple components (multi-agent LLM method and diffusion-based time-series generation), where each component itself has many details. However, in this situation I believe the authors need to do a better job illustrating their method using figures, specific language/terms, removing jargon, and potentially streamlining some details or moving them to the appendix.

I believe this work can benefit most from a better interaction between the text and the figures. In it's current state, I believe this manuscript does not fully utilize the figures. I would like to see a subfigure/box/illustration for each key component which is referenced in the text. This way a reader can follow along visually and via the text. This is very helpful because there are a lot of moving parts in this paper and it is easy to forget the relationship between these parts. In a well made figure that is integrated into the text, the reader can look at the figure to remind themselves how each component fits into the overall scheme. Practically, it would be good if the authors refer to a figure in every or everyother sentence in sections 3 and 4.

- There is a lot of terms and jargon that I believe could be better defined or removed. Figures will also help with this.
- The abstract says this work outperform on 10 out of 12 datasets, but the introduction states 11 out of 12.
- Please run this work through a spell/grammer check, I found misspelled words and missing punctuation.

# Results
Figure 3 is both very hard to read and poorly made. Both of these lead to a convoluted message and potentially erroneous interpretations. For example, to me it looks like the in (2) the trend stays the same, but the diffusion model only introduces extra noise that varies with conditioning. However, this is not the entirely correct interpretation.
- Increase font size of the axis
- merge the x axes in each column
- All plots in each column need to have the same y range
- Fill the white space to the left and right

From this Figure 3, it seems that generated time series are much noisier. I would like to see how the overall variance of the datasets compare from original, without conditioning, and with conditioning, along with some (possibly brief) discussion.

**Questions:**

1) In what ways do the LLM prompts succeed and fail?
2) In what ways do the text succeed and fail to describe the time series (e.g. sinusoidal, trending upward, extreme values, ...)?

---

> ### Author Response · Authors · 2024-11-22
> **Kindly reply**
>
> Thank you for your kind advice. We really appreciate your suggestion and already revise them in the new version of papers.
>
> **❓ In what ways do the LLM prompts succeed and fail?**
>
> 💡  First, it is difficult for LLMs to strictly adhere to the given prompt. This also applies to LLMTime ([[2310.07820] Large Language Models Are Zero-Shot Time Series Forecasters](https://arxiv.org/abs/2310.07820)). This shows that directly using LLM as backbone is still an open challenge. Second, similar to the findings of Chenglei’s work ([[2210.09150] Prompting GPT-3 To Be Reliable](https://arxiv.org/abs/2210.09150)), excessive contextual information is more often detrimental than beneficial for prompting LLMs at this stage, as it makes it difficult for the model to extract key information, thereby reducing output quality. As shown in Table 1, the initial detailed descriptions actually misled the LLM. Third, Although aligning multimodal information has been shown to be effective ([Image-based time series forecasting: A deep convolutional neural network approach - ScienceDirect](https://www.sciencedirect.com/science/article/abs/pii/S0893608022003902)), directly aligning TS or decomposed TS to LLM does not significantly improve the performance, as shown in Appendix A.
>
> **❓In what ways do the text succeed and fail to describe the time series (e.g. sinusoidal, trending upward, extreme values, ...)?**
>
> 💡  Here is a summary about the influence of different text components
>
> - Background information on the data is useful: this may be because the LLM encountered related information during the pre-training stage.
> - Mean, maximum, minimum, and variance are useful: these define the range of values.
> - Overall trend is useful: this can help prevent the LLM from fluctuating within a specific range.
> - The length of the generation is very useful: this can be a good way to limit LLM from generating too short time series.
> - Specific trends are not useful: the LLM is still unable to understand subtle differences between time steps.
>
> **Comments, Suggestions And Typos:** We appreciate your detailed feedback and the constructive suggestions. We have carefully follow your advice to revised paper in order to improve the clarity and readability. To address the concerns about comprehensibility, we have:
>
> - We rephrase the section 3 and section 4 to streamline explanations and use more specific language to clearly describe each component of our method. More Detail could be find in Appendix A.3, A.4 and A.5
> - We redraw the Figure 1, 2 to provide clearly describe of our methods. We keep cite each of figure to explain the sentence to help reader follow the interactions between components more easily and understand the overall scheme at a glance.
> - In addition, we have clarified terminology and performed a thorough grammar and spell check to correct any errors. We also addressed the discrepancy in the results mentioned in the abstract and introduction, now ensuring consistency between sections.
>
> Since none of the comments outline serious issues with the papers’ methodology or findnigs, please give us the opportunity to be present in the proceedings.

---

> > ### Author Response · Authors · 2024-11-25
> >
> > Thank you once again for your thoughtful review and valuable feedback! As we approach the end of the discussion period, we want to ensure that our previous responses have fully addressed all your concerns. If you have any additional questions or unresolved issues that we can clarify to achieve a better score, please don’t hesitate to let us know. We’re more than happy to assist further!

---

### Author Response · Authors · 2024-11-22
**To All**

We thank all reviewers for their hard work and thoughtful suggestions. We have addressed all the issues mentioned and incorporate further improvements in revised paper. The revised version of the paper has been uploaded as a new revision on OpenReview. We highlight all the change with different color. Additionally,  the code can be accessed via the anonymous GitHub link provided https://anonymous.4open.science/r/BRIDGE-CB58

For your convenience, here is a TL;DR summary of all the changes made in the revised version of the paper

**Introduction**

1. We corrected all grammatical errors.
2. We adjusted our statement to clear our motivation that the proposed model contributes to using text to provide context information for cross-domain time series generation instead of building a new foundation model for time series forecasting.
3. We rectify the statement in our summary paragraph from “11 of 12” to “10 of 12”.

**Related Work**

1. We condensed the write-up to allow more room for discussing methodology and experiments.

**Methodology**

1. We condensed the overview of the section 3.
2. We redrew the model in **Figure 1 and Figure 2** for better understanding.
3. We have rewritten **Subsection 3.2, Step 1**, to include a clearer explanation of the proposed single-agent text template system. We have kept the description concise in the main body of the paper and moved the system pipeline to **Appendix A.2**, and the text template to **Appendix A.3**.
4. We have rewritten **Subsection 3.2, Step 2** for clear how evaluation works.
5. We have rewritten **Subsection 3.3 Step 3** and moved additional details about structure and workflow to **Appendix A.5**; Iteration generation sample in **Appendix A.6**; And model workflow example in **Appendix A.7.**

**Result Analysis**

1. We added an analysis of the impact of different strategies of the multi-agent system in **Subsection 6.1** with **Table 1**.
2. We added a diffusion-based model as baseline in **Table 3**.
3, We added Chronos's KernelSynth data augmentation methods in **Table 4**.
4. We revised **Subsection 6.5** to add more experiments and analysis of prototypes, including visualizing the output in newly made **Figures 4**, **5** and **6.**

**Appendix**

1. We added a text template example in the **Appendix A.3**
2. We added a text sample examples for each iteration in **Appendix A.4**.
3. We added an example output of the collaboration between multiple agents in **Appendix A.5**’s **Table 7, 8, 9** and **10.**
4. We added more hyper-parameters and training details in **Appendix B.2**
5. We added a detailed model architecture description in **Appendix B.3**
6. We moved the description of the experiment setup to **Appendix D**
7. We added the experimental results of using different text models in **Appendix H**
8. We add the visualization samples corresponding to prototypes in different domains in **Appendix J**

---

### Author Response · Authors · 2024-11-30
**To all v2**

We sincerely thank the reviewers for their thoughtful feedback and valuable suggestions, which have greatly helped us refine and strengthen our work. We appreciate the time and effort you have dedicated to reviewing our submission. Your insights have allowed us to clarify key aspects of our approach and results.

If there are any additional questions or points requiring further clarification, we would be delighted to address them to ensure our work meets the highest standards. We hope these updates provide a clearer understanding of our contributions and improve the overall evaluation of our submission.

So far, we have addressed the majority of the concerns raised. Below is a brief summary of the actions we have taken:

✅ Multiple runs and average performance reporting, as requested by **bfZV**

✅ Different strategies of agents, as requested by **x5xf** and **ZPw4**

✅ Prototypes visualization and analysis, as requested by **ZPw4**

✅ Addition of a diffusion model-based baseline, as requested by **ZPw4**

✅ Use of different text models, as requested by **ZPw4**

✅ Exploration of different model mechanism settings, as requested by **ZPw4**

✅ Use of Chronos’s DA as a baseline, as requested by **1C8B**

In the latest revised version of our paper, we have added the results of the diffusion baseline model and included a discussion of its performance in the main text. We have also compared different data augmentation methods on the ILI and M4 datasets, which are reflected in the following:

1. A diffusion-based model as a baseline, added in **Table 3**
2. Chronos data augmentation methods, added in **Subsection 6.4** and **Appendix I** (**Tables 13 and 14**)

We will include additional experimental results discussed during the rebuttal phase and correct any inappropriate descriptions in the next version of the paper. We kindly request the opportunity to present our work in the proceedings.

---

### Meta-Review · Area_Chair_QsrD · 2024-12-11

**Metareview:**

The paper proposes a new method for generating synthetic time series data from textual descriptions. This involves a multi-agent LLM framework to refine text descriptions and a novel diffusion model (BRIDGE) conditioned on these descriptions.  The authors claim this approach improves cross-domain time series generation, particularly for low-resource domains,

Strengths of the Paper:

Novelty: The paper tackles a relatively new problem in time series analysis - building high-quality datasets with text components.
Multi-agent Approach: The use of a multi-agent system for refining textual descriptions is innovative and shows promising results.
Diffusion Model: The proposed diffusion-based method generates high-quality time series data with diverse properties.
Comprehensive Evaluation: The authors test various aspects of their method, including the impact of different text components and the quality of generated data for model training.

Weaknesses of the Paper:

Misleading Claims: The paper's claims about resource efficiency compared to prior work (Chronos) seem misleading. The authors do not adequately address this concern.
Potential Data Leakage: The web scraping approach used for generating text descriptions raises concerns about potential data leakage. The authors' response does not convincingly demonstrate that this is avoided.
Lack of Clarity: The paper lacks important details about the agent design, diffusion model specifics, and prototype analysis.
Experimental Concerns: The experimental setup has weaknesses, including not comparing to diffusion model baselines and using pre-trained models, which introduce confounding factors.

Despite some strengths, the paper's weaknesses justify rejection. The misleading claims about resource efficiency and the potential data leakage raise concerns about the validity of the results.

**Additional Comments On Reviewer Discussion:**

Summary of Discussion and Changes during Rebuttal:

The rebuttal period involved addressing reviewers' concerns about clarity, experimental design, and missing details.

Key Points Raised by Reviewers:

Tart: Pointed out writing issues and suggested improving figure quality.
bFZV: Requested clarification on the performance of BRIDGE without text and asked for multiple runs and standard deviation reporting.
1C8B: Raised concerns about misleading comparison to Chronos, potential data leakage, and lack of details.
ZPw4: Asked for more details on agent design, diffusion model specifics, and prototype analysis.
x5xf: Requested code and data examples, and questioned the impact of removing parts of the agent architecture.
How the Authors Addressed These Points:

The authors revised the paper to improve writing and figure quality.
They provided explanations for the performance of BRIDGE without text and reported averaged results with standard deviation.
They attempted to clarify the comparison to Chronos and address the data leakage concern, but not convincingly.
They provided more details on agent design, diffusion model specifics, and prototype analysis.
They released code and provided data examples, and analyzed the impact of removing parts of the agent architecture.

Weighing in Each Point in the Final Decision:

While the authors attempted to address the reviewers' concerns, they did not fully resolve the core issues. The clarifications regarding resource efficiency and data leakage were not convincing, and the experimental concerns remained. The improvements in writing, figure quality, and details are appreciated, but they do not outweigh the fundamental weaknesses of the paper.

---

### Decision · Program_Chairs · 2025-01-22

Reject